taxonomy and systematics

*Acanthechiniscus*, cuticle, *Multipseudechiniscus*, phylogenetic congruence, phylogeny, systematics

**Authors for correspondence:**
Piotr Gąsiorek
e-mail: piotr.lukas.gasiorek@gmail.com
Łukasz Michalczyk
e-mail: LM@tardigrada.net

# Revised *Cornechiniscus* (Heterotardigrada) and new phylogenetic analyses negate echiniscid subfamilies and tribes

## Piotr Gąsiorek and Łukasz Michalczyk

Institute of Zoology and Biomedical Research, Jagiellonian University, Gronostajowa 9, 30-387 Kraków, Poland

PG, 0000-0002-2814-8117; ŁM, 0000-0002-2912-4870

Echiniscidae are undoubtedly the most thoroughly studied lineage of the class Heterotardigrada. Recently, new subfamilies and tribes grouping echiniscid genera based on traditionally recognized morphological clues have been proposed. Here, by integrative analyses of morphology and DNA sequences of numerous populations of a rare genus *Cornechiniscus*, we show that this hypothesized classification is artificial. Specifically, we demonstrate that Echiniscinae are diphyletic, as *Bryodelphax* forms a distinct phyletic lineage within Echiniscidae, and Pseudechiniscinae are polyphyletic, with *Mopsechiniscus* being indirectly related to *Pseudechiniscus*, which is closer to the *Echiniscus*-like genera than to other genera with pseudosegmental plates. Consequently, the subfamilies and tribes are considered as unsupported from the phylogenetic and morphological point of view. The genus *Cornechiniscus* is revised, and the phenotypic diagnoses of several species are updated thanks to new rich material from Africa, Asia and Europe. *Cornechiniscus imperfectus* sp. nov. is described from mountains of Kyrgyzstan, being the second appendaged species within the genus and the third known to exhibit dioecy. A taxonomic key to the genus is provided. Systematic positions of *Acanthechiniscus* and *Multipseudechiniscus* are also discussed. *Acanthechiniscus goedeni* is confirmed to be a member of the genus *Acanthechiniscus*.

## 1. Introduction

Tardigrades belong to the superclade of moulting animals, the Ecdysozoa [1,2]. The level of understanding of phylogenetic relationships within tardigrade lineages is diversified, as some

clades have relatively well-resolved affinities (e.g. some macrobiotoids; [3–5]), whereas others remain obscure, primarily due to the rare occurrence and undersampling (e.g. many arthrotardigrade families; [6]). The family Echiniscidae, although the most extensively studied limnoterrestrial lineage of predominantly marine Heterotardigrada since the times of early tardigradologists [7–9], are far from reaching a stable systematic arrangement of genera. Despite the continuous effort in obtaining novel genetic and morphological data for echiniscid species, resulting in the establishment of new genera, such as *Multipseudechiniscus* [10], *Diploechiniscus* [11], *Acanthechiniscus* [5] and *Stellariscus* [12], and improved comprehension of morphological diversity within the already established genera [12–18], echiniscid phyletic relationships are not well resolved [19]. There are two echiniscid clades commonly accepted as monophyletic and supported in various phylogenetic analyses: the *Echiniscus* lineage (((*Hypechiniscus* (*Testechiniscus* (*Diploechiniscus* + *Echiniscus*-like genera))) and the clade comprising ((*Proechiniscus* (*Acanthechiniscus* + *Cornechiniscus*)) [5,12,20]. The relationships of the remaining genera are debatable [21]. Recently, a new hypothetical grouping of the genera into subfamilies and tribes, founded primarily on the older schemes of natural history of the Echiniscidae [22,23], was proposed by Guil *et al.* [24]. Specifically, the subdivision and arrangement of dorsal plates were used as putative synapomorphies for these taxonomic ranks, although molecular phylogenies presented therein did not support some (Pseudechiniscinae), or lacked genetic information for certain genera (e.g. *Multipseudechiniscus*, *Novechiniscus*).

This work offers an extensive revision of the genus *Cornechiniscus* [25]. Type specimens, unpublished museum materials, embracing new records for different countries, and newly obtained populations from Ethiopia, France and Kyrgyzstan allowed for an amendment of the descriptions for nine known species and a description of *Cornechiniscus imperfectus* sp. nov. from Central Asia. The morphological patterns within the genus and phyletic affinities of its members are elucidated with the support of genetic data (four DNA markers), light contrast microscopy (LCM) and scanning electron microscopy (SEM) imaging. An updated taxonomic key to all currently recognized species of the genus is also provided. Essentially, the influx of new sequences has contributed to the re-arrangement of genera on the familial tree of Echiniscidae, calling into question the morphology-based subfamilial and tribal classification recently proposed by Guil *et al.* [24]. Finally, synapomorphies for the clade ((*Proechiniscus* (*Acanthechiniscus* + *Cornechiniscus*)) + likely *Multipseudechiniscus* are proposed, and *Acanthechiniscus goedeni*, of which the generic affinity was uncertain [5], is ascertained herein as the member of the genus *Acanthechiniscus*.

# 2. Material and methods

## 2.1. Sample processing and comparative material

Tardigrades were isolated from moss and lichen samples intermingled with high amounts of soil, collected by numerous people (see table 1 for details), and processed following the protocol described by Stec *et al.* [26]. Type materials of *C. brachycornutus* [27], *C. ceratophorus* [28], *C. holmeni* [29], *C. lobatus* [30], *C. madagascariensis* [31], *C. schrammi* [32], *C. subcornutus* [25], *C. tibetanus* [33], *Multipseudechiniscus raneyi* [34] and *Acanthechiniscus goedeni* [34] deposited in the Natural History Museum of Verona, Italy and the Natural History Museum of Denmark were examined using LCM. Moreover, slides with *Cornechiniscus* spp. representing unpublished records, deposited in both institutions, were also studied.

## 2.2. Microscopy and imaging

Specimens for light microscopy and morphometry were mounted on microscope slides in Hoyer's medium according to Morek *et al.* [35] and, together with the material cited above, examined under a Nikon Eclipse 50i phase-contrast microscope (PCM) fitted with a Nikon Digital Sight DS-L2 digital camera. Specimens for imaging in SEM were prepared according to Stec *et al.* [26]. Bucco-pharyngeal apparatuses were extracted following the sodium hypochlorite protocol provided by Eibye-Jacobsen [36] with modifications described in Gąsiorek *et al.* [37]. Both animals and apparatuses were examined under high vacuum in a Versa 3D DualBeam SEM at the ATOMIN facility of Jagiellonian University, Kraków, Poland. For deep structures that could not be fully focused in a single photograph, a series of one to three images were taken every *ca* 0.1 µm and then assembled with Corel into a single deep-focus image.

**Table 1.** Collection data for populations of different species of *Cornechiniscus* investigated in this study. Types of analyses: LCM—imaging and morphometry in PCM/NCM, imaging in SEM, DNA sequencing. Number in each analysis indicates how many specimens were used in a given method. Meaning of symbols: a, adult females; v, exuviae; j, juveniles; l, larvae.

| species | sample code | locality | coordinates and altitude | sample type and environment | accompanying echiniscid species | collector | analyses LCM | SEM | DNA |
|---|---|---|---|---|---|---|---|---|---|
| *C. cornutus* | KG.123 | Kyrgyzstan, Issyk-Kul province, Tong District | 42°02'51.6'' N 77°00'35.8'' E 2175 m.a.s.l. | moss mixed with soil on rock, mountains | *Testechiniscus spitsbergensis* | B. Surmacz, W. Morek | 6a | — | — |
| | KG.127 | Kyrgyzstan, Naryn province, Kum-Döbö | 42°13'22.7'' N 75°27'17.4'' E 2023 m.a.s.l. | moss mixed with soil, mountains | *Echiniscus testudo* | B. Surmacz, W. Morek | 2a | — | 4a |
| | KG.128 | Kyrgyzstan, Naryn province, Kum-Döbö | 42°13'22.6'' N 75°27'17.8'' E 2034 m.a.s.l. | moss mixed with soil, mountains | *Echiniscus testudo* | B. Surmacz, W. Morek | 5a | — | 2a |
| *C. imperfectus* sp. nov. | KG.012 | Kyrgyzstan, Jalalabat province, vicinity of Tashkömür | 41°22'22.2'' N 72°1'43.0'' E 726 m.a.s.l. | moss + lichen mixed with soil, mountains | *Echiniscus testudo* | B. Surmacz, W. Morek | 30a + 5j + 1l | 35a | 1a |
| | KG.013 | Kyrgyzstan, Jalalabat province, vicinity of Tashkömür | 41°22'22.5'' N 72°1'46.6'' E 762 m.a.s.l. | moss + lichen mixed with soil, mountains | *Echiniscus testudo* | B. Surmacz, W. Morek | 5a + 1j + 2v | — | 4a |
| | KG.014 | Kyrgyzstan, Jalalabat province, vicinity of Tashkömür | 41°22'23.2'' N 72°1'48.3'' E 784 m.a.s.l. | moss + lichen mixed with soil, mountains | *Echiniscus testudo* | B. Surmacz, W. Morek | 19a + 2j + 2l | — | 1a |

**Table 1.** (Continued.)

| species | sample code | locality | coordinates and altitude | sample type and environment | accompanying echiniscid species | collector | analyses | | |
|---|---|---|---|---|---|---|---|---|---|
| | | | | | | | LCM | SEM | DNA |
| C. lobatus[a] | FR.139 | France, Saint-Maur-des-Fossés | 48°48'33.4'' N 2°29'07.1'' E 46 m.a.s.l. | moss, urban cemetery | — | W. Morek | 18a + 17l + 3v | 10a | 4a |
| | KG.115 | Kyrgyzstan, Issyk-Kul province, Tong District | 42°02'51.7'' N 77°00'37.0'' E 2151 m.a.s.l. | moss mixed with soil, mountains | Testechiniscus spitsbergensis | B. Surmacz, W. Morek | 8a | — | — |
| | KG.118 | Kyrgyzstan, Issyk-Kul province, Tong District | 42°02'52.0'' N 77°00'36.3'' E 2157 m.a.s.l. | moss + lichen mixed with soil, mountains | Echiniscus testudo | B. Surmacz, W. Morek | 2a | — | — |
| C. madagascariensis | ET.007 | Ethiopia, Amhara Regional State, Cusquam ruins | 12°37' N 37°27' E 2200 m.a.s.l. | moss mixed with soil on rock, rural | — | A. K. Pedersen | 13a + 9j + 2l + 5v | 10a | 8a |
| C. subcornutus | KG.129 | Kyrgyzstan, Jalalabat province, Toluk | 41°55'12.6'' N 73°38'00.1'' E 1463 m.a.s.l. | moss mixed with soil on rock, mountains | Echiniscus testudo | B. Surmacz, W. Morek | 19a + 4j | 6a | 4a |
| | KG.144 | Kyrgyzstan, Jalalabat province, Toluk | 41°55'04.9'' N 73°37'50.5'' E 1517 m.a.s.l. | moss mixed with soil on rock, mountains | Echiniscus trisetosus | B. Surmacz, W. Morek | 3a | — | — |

[a]Also found in large numbers in the samples: KG.012 (3a), KG.013 (66a + 4l), KG.014 (1a), KG.127 (1a), KG.128 (25a).

**Table 2.** Primers and references for specific protocols for amplification of the four DNA fragments sequenced in the study.

| DNA fragment | primer name | primer direction | primer sequence (5′-3′) | primer source | PCR program[a] |
|---|---|---|---|---|---|
| 18S rRNA | 18S_Tar_Ff1 | forward | AGGCGAAACCGCGAATGGCTC | [43] | [48] |
| | 18S_Tar_Rr2 | reverse | CTGATCGCCTTCGAACCTCTAACTTTCG | [40] | |
| 28S rRNA | 28S_Eutar_F | forward | ACCCGCTGAACTTAAGCATAT | [44] | [45] |
| | 28SR0990 | reverse | CCTTGGTCCGTGTTTCAAGAC | [45] | |
| ITS-1 | ITS1_Echi_F | forward | CCGTCGCTACTACCGATTGG | [46] | [49] |
| | ITS1_Echi_R | reverse | GTTCAGAAAACCCTGCAATTCACG | [46] | |
| COI | bcdF01 | forward | CATTTTCHACTAAYCATAARGATATTGG | [47] | [49] |
| | bcdR04 | reverse | TATAAACYTCDGGATGNCCAAAAAA | [47] | |

[a]All PCR programs are also provided in Stec et al. [26].

## 2.3. Morphometrics and terminology

All measurements are given in micrometres (μm) and were performed under PCM. Structures were measured only if not damaged and their orientations were suitable. Body length was measured from the anterior to the posterior end of the body, excluding the hind legs. The *sp* ratio is the ratio of the length of a given structure to the length of the scapular plate ([38]; values presented in italics throughout the text and tables). Morphometric data were handled using the Echiniscoidea v. 1.2 template available from the Tardigrada Register, www.tardigrada.net/register [39]. The terminology follows Kristensen [22], with subsequent changes proposed in Gąsiorek et al. [20,40]. Body appendage formula follows Marcus [41] and Gąsiorek et al. [40]. The *ps* signifies appendages present on the posterior margin of the pseudosegmental plate IV′. In the genus *Cornechiniscus*, cirri *A* are horn-shaped, but the traditional naming is preserved for nomenclatural consistency among echiniscids. Raw morphometric data for the new species are deposited in the Tardigrada Register under www.tardigrada.net/register/0068.htm and in the electronic supplementary material.

## 2.4. Genotyping

DNA was extracted from individual animals using Chelex® 100 resin [42]. We sequenced four DNA fragments: a small ribosome subunit (18S rRNA), a large ribosome subunit (28S rRNA), an internal transcribed spacer I (ITS-1) and the subunit I of cytochrome c oxidase (COI). All fragments were amplified and sequenced according to the protocols described by Stec et al. [26], and the primers and specific PCR programs can be found in table 2. Sequencing products were read with the ABI 3130xl sequencer at the Molecular Ecology Laboratory of the Institute of Environmental Sciences at Jagiellonian University. Sequences were processed using v. 7.2.5 of BioEdit [50] and submitted to GenBank: 18S rRNA (MT420868-73), 28S rRNA (MT420852-7), ITS-1 (MT420858-67), COI (MT420437-57). Paragenophores and/or hologenophores were mounted on microscope slides in Hoyer's medium [51] and are deposited in the Jagiellonian University.

## 2.5. Phylogenetics

We aligned all available echiniscid 18S rRNA and 28S rRNA sequences deposited in GenBank (excluding 28S rRNA sequences from Jørgensen et al. [19] as they represent a non-homologous fragment of this marker), using the Q-INS-I strategy which considers the secondary structure of RNA in MAFFT v. 7 [52,53], to reconstruct the phylogeny of Echiniscidae. *Oreella mollis* [54] was chosen as the outgroup in this analysis. The aligned fragments were edited and checked manually in BioEdit and later concatenated (datasets included in the electronic supplementary material). Using PartitionFinder v. 2.1.1 [55] under the Akaike information criterion (AIC) and *greedy* algorithm [56], the best substitution model and partitioning scheme were chosen for posterior BI phylogenetic analysis. As best-fit partitioning scheme, PartitionFinder suggested to retain two predefined partitions separately, and for each, the best-fit model was GTR + I + G. ModelFinder [57] under the AIC and corrected AIC

(AICc) was used to find the best substitution models for two predefined partitions in the case of ML analysis [58]. The program indicated following models: SYM + I + G4 (18S rRNA) and TVMe + I + G4 (28S rRNA). In order to elucidate the relationships between the five analysed species of *Cornechiniscus*, all obtained sequences of ITS-1 and COI from single animals underwent an identical procedure, but final substitution models were different than for the concatenated 18S + 28S rRNA dataset: (i) Bayesian inference (BI): the first partition GTR + G, the second partition K81UF + I + G, the third partition HKY + I + G and the fourth partition GTR + I; (ii) maximum likelihood (ML): the first partition TIM2 + F + G4, the second partition K3Pu + F + I, the third partition HKY + F and the fourth partition K2P + G4. *Echiniscus succineus* [59] was chosen as the outgroup in this case. Phylogenetic trees produced using single DNA markers (i.e. either ITS-1 or COI) gave the same topologies as the tree inferred from two concatenated variable markers, thus the tree presented herein is based on the concatenated dataset.

ML topologies were constructed using IQ-TREE [60,61]. In ML, support for internal nodes was measured using 1000 ultrafast bootstrap replicates [62]. Bootstrap (BS) support values greater than or equal to 70% in the final tree were regarded as significant statistical support. BI marginal posterior probabilities were calculated using MrBayes v. 3.2 [63]. Random starting trees were used and the analysis was run for 10 million generations, sampling the Markov chain every 1000 generations. An average standard deviation of split frequencies of less than 0.01 was used as a guide to ensure the two independent analyses had converged. The program Tracer v. 1.3 [64] was then used to ensure Markov chains had reached stationarity and to determine the correct 'burn-in' for the analysis, which was the first 10% of generations. The effective sample size (ESS) values were greater than 200 and consensus tree was obtained after summarizing the resulting topologies and discarding the 'burn-in'. The BI consensus tree clades recovered with posterior probability (PP) between 0.95 and 1.00 were considered well supported, those with PP between 0.90 and 0.94 were considered moderately supported and those with lower PP were considered unsupported. All final consensus trees were visualized in FigTree v. 1.4.3 available from http://tree.bio.ed.ac.uk/software/figtree.

# 3. Results

## 3.1. Phylogeny of Echiniscidae and *Cornechiniscus*

The BI and ML phylogenetic analyses of the family Echiniscidae based on conservative nuclear markers, 18S and 28S rRNA, showed that *Bryodelphax* was the sister genus to all other echiniscids (figure 1). The remaining four lineages stayed in a polytomy: *Parechiniscus*, *Mopsechiniscus*, the *Cornechiniscus* clade ((*Proechiniscus* (*Acanthechiniscus* + *Cornechiniscus*)) and *Pseudechiniscus* + the *Echiniscus* clade comprising *Hypechiniscus*, *Testechiniscus*, *Diploechiniscus*, *Echiniscus*, *Barbaria*, *Claxtonia*, *Kristenseniscus*, *Nebularmis* and *Viridiscus*.

The BI and ML phylogenetic analyses of the genus *Cornechiniscus* based on the more variable markers, ITS-1 and COI (figure 2), showed that *C. subcornutus* was the sister species to the other four analysed *Cornechiniscus* spp., which grouped into two sister clades, one comprising *C. cornutus* + *C. imperfectus* sp. nov. and the other—*C. lobatus* + *C. madagascariensis*. The *C. lobatus* clade exhibited a poorly differentiated genetic structure with the single European population embedded between the Asian populations.

## 3.2. Amendments to *Cornechiniscus* spp. descriptions and the description of *Cornechiniscus imperfectus* sp. nov.

### 3.2.1. *Cornechiniscus brachycornutus*

*Cirri A* reduced, with indistinguishable cirrophores. Sculpture evident and consisting of endocuticular pillars protruding through epicuticle as granules of two different sizes: larger on the central portions of the cephalic, scapular, median, paired segmental plates and posterior portion of the caudal plate, and smaller on the lateral portions, the anterior portion of the caudal plate and on the pseudosegmental plate (figure 3a). Venter with deep cuticular grooves. Dentate collar on legs IV reduced, with two short teeth [27]. Claws heteronych (anisonych), i.e. claws IV slightly longer than claws I–III, sometimes with spurs directed upwards.

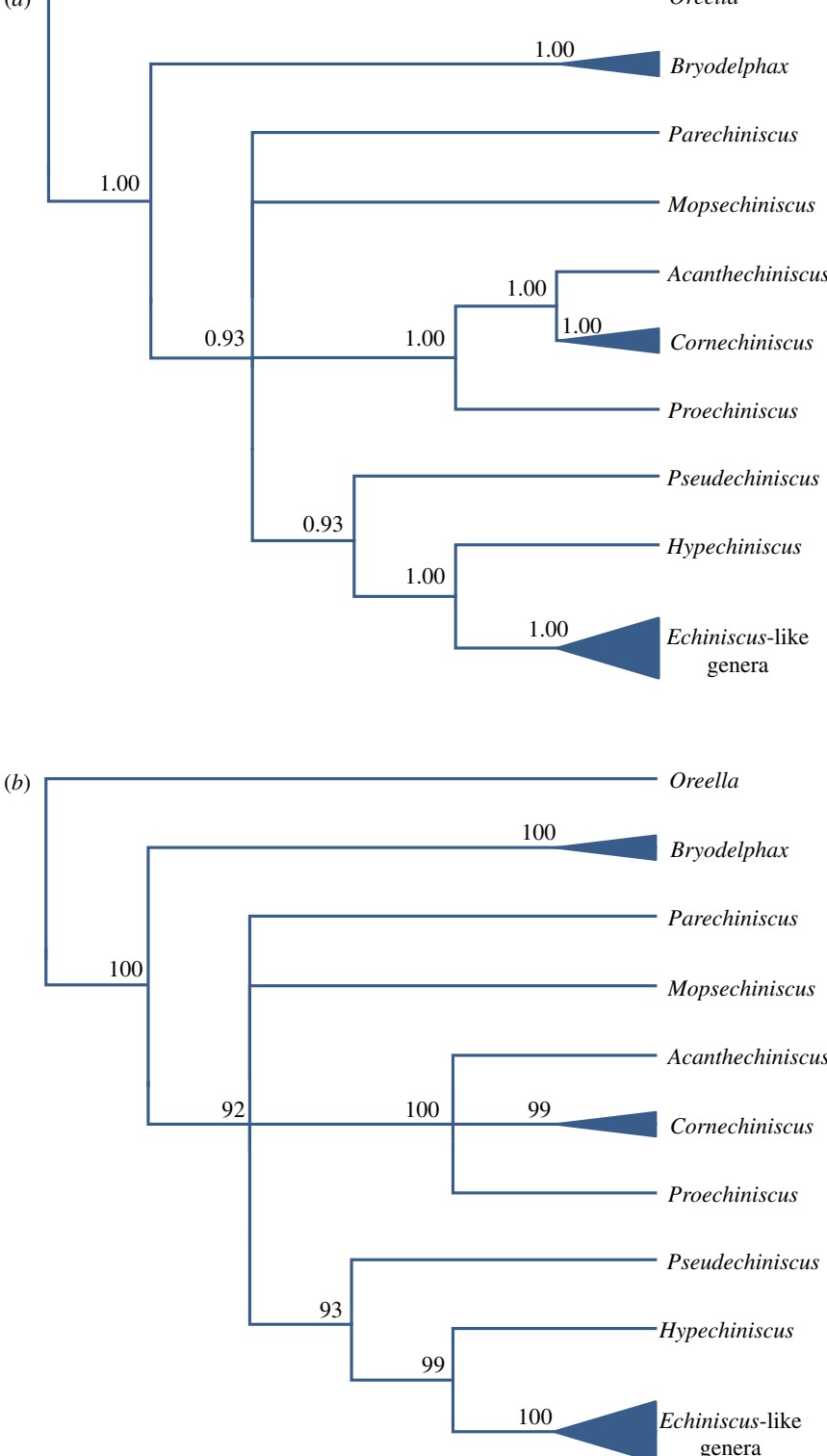

**Figure 1.** (*a*) The Bayesian reconstruction (cladogram) of phylogenetic relationships of Echiniscidae based on concatenated 18S + 28S rRNA sequences, with *Oreella mollis* used as an outgroup. Branch support is given as Bayesian PP values above branches. Branches with support below 0.90 in BI were considered unsupported and are not shown on the presented tree. (*b*) The maximum-likelihood reconstruction (cladogram) of phylogenetic relationships of Echiniscidae based on concatenated 18S + 28S rRNA sequences, with *Oreella mollis* used as an outgroup. Branch support is given as bootstrap values above branches. Branches with support below 70% were considered unsupported and are not shown on the presented tree.

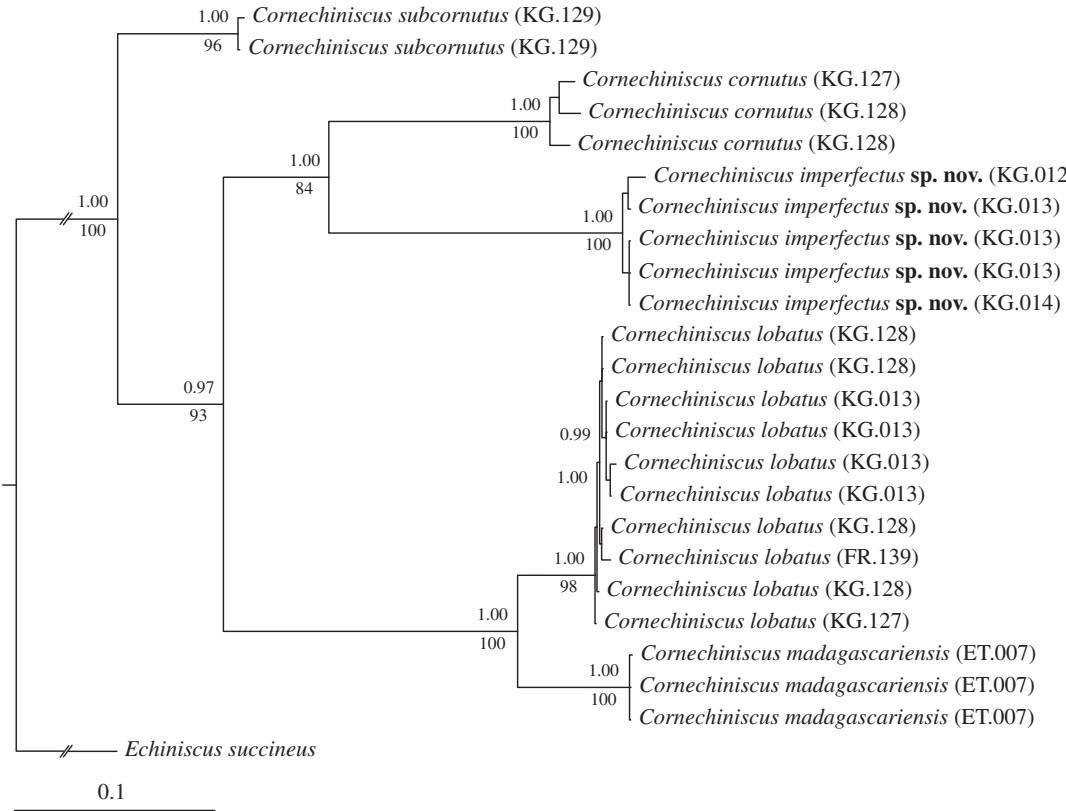

**Figure 2.** The phylogenetic relationships within *Cornechiniscus* based on concatenated COI + ITS-1 sequences, with *Echiniscus succineus* used as an outgroup. Branch support is given as Bayesian posterior probability values above branches, and bootstrap values (maximum-likelihood) below branches. The scale bar refers to the Bayesian tree.

### 3.2.2. Cornechiniscus cornutus

*Cirri A* typical, with well-delimited, lighter cirrophores. *Cirri A* composed of two evident layers of cuticular material differing in density: the less dense and hence brighter outer sheath, and the inner core, denser and consequently darker in PCM (figure 3*b*). Dorsal plates poorly marked, with most sclerotized posterior margins of the cervical, scapular and paired segmental plates. Borders of plates visible mostly as darker lines in PCM, being more prominent extensions of epicuticular layer. Sculpture faint and minute, with largest endocuticular pillars on anterior portions of median plates 1–2. Venter with deep cuticular grooves. Dentate collar on legs IV absent. Claws miniaturized and isonych, i.e. claws I–IV equal or very similar in length, always spurless.

### 3.2.3. Cornechiniscus ceratophorus

*Cirri A* elongated and slender, with well-delimited, lighter cirrophores. Margins of all dorsal plates strongly sclerotized, especially on the posterior margins of the scapular, paired segmental and pseudosegmental plates in the contact zone with dorsal spines $B^d$, $C^d$, $D^d$ and *ps*. Sculpture evident and atypical for the genus, consisting of almost ideally circular endocuticular pillars [28]. Venter with deep cuticular grooves. Dentate collar on legs IV also atypical for the genus, i.e. not reduced and consisting of 2–7 long, acute teeth [28]. Claws slightly heteronych (anisonych), that is with claws IV longer than claws I–III, with basal spurs directed upwards.

### 3.2.4. Cornechiniscus holmeni

*Cirri A* reduced, with indistinguishable cirrophores. Body appendage formula A-C-D-($D^d$)-(*ps*)-E. Lateral appendages *C* and *D* in the form of long, filamentous cirri, whereas appendages $D^d$, *ps* and *E* developed as robust spines, of which spines *E* are the longest (table 3). Asymmetry in appendage configuration

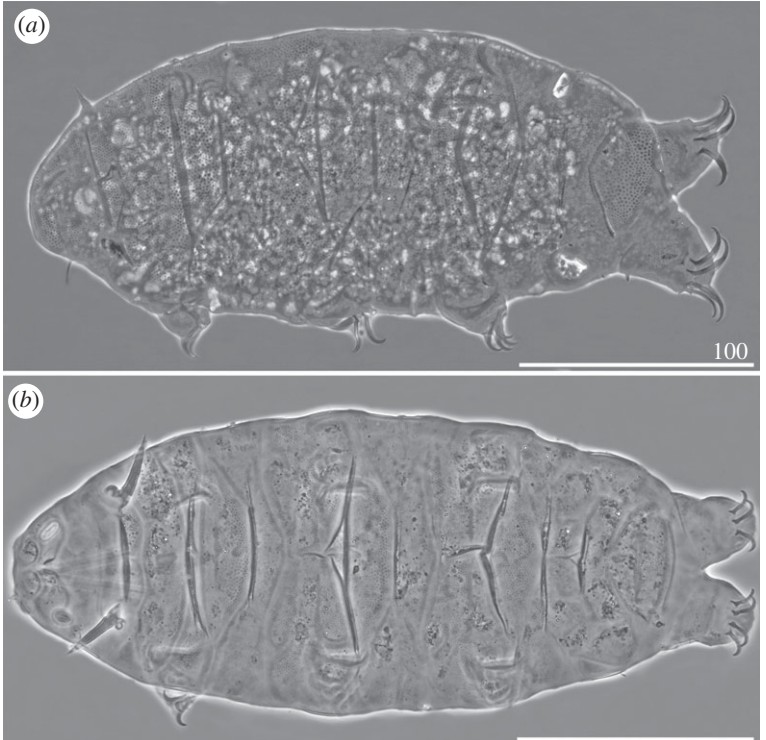

**Figure 3.** Habitus of females (PCM): (*a*) *Cornechiniscus brachycornutus* (dorsolateral view, holotype); (*b*) *Cornechiniscus cornutus* (dorsal view, specimen from Kyrgyzstan). All scale bars = 100 µm.

infrequent, typically occurring in dorsal spines $D^d$ and *ps*. Spines $D^d$ rarely absent. Margins of all dorsal plates strongly sclerotized, especially on the posterior margins of the scapular, second paired segmental, pseudosegmental plates and caudal incisions in the contact zone with dorsal spines $D^d$, *ps* and *E* (figure 4*a*). Sculpture consisting of minute, densely arranged endocuticular pillars most discernible on the cephalic and caudal (terminal) plates (figure 4*b*). Venter smooth. Spine I and the single spine constituting dentate collar IV long, strong and acute (figure 4*a*). Claws strongly heteronych, spurless or with asymmetric spurs pointing upwards.

Larvae with already developed adult morphology (figure 4*b*), except for the lack of the gonopore and anus. Moreover, larvae exhibit even stronger heteronych claws than adults (compare *sp* values for claws I–III and claws IV of adult females and larvae; tables 3 and 4).

### 3.2.5. *Cornechiniscus imperfectus* sp. nov.

urn:lsid:zoobank.org:act:4727E334-875D-4F17-A1B6-F8F35278D377

Type material: Holotype (adult female on the slide KG.012.07), allotype (adult male on the slide KG.013.06) and 63 paratypes (33 females, 19 males, eight juveniles and three larvae; slides KG.012.02–23, KG.013.02, 06, 07, 10, KG.014.01–12). Further 20 paratypes on the SEM stub no. 17.02. With the exception of following slide numbers: KG.013.10 (female and juvenile; Department of Animal Biology, Catania collection, Italy), KG.014.08 (two females; the University of Modena and Reggio Emilia, Italy), KG.014.09 (female and male; Comenius University in Bratislava, Slovakia), KG.014.10–12 (five females, four males and juvenile; Natural History Museum of Denmark, University of Copenhagen), type specimens deposited in the Faculty of Biology, Jagiellonian University, Poland.

*Locus typicus*: *ca* 41°22′22″ N, 72°14′43″ E; 720–790 m.a.s.l.; Kyrgyzstan, Jalalabat province, vicinity of Tashkömür; moss mixed with soil.

Description of the new species:

*Females* (i.e. from the third instar onwards; measurements and statistics in table 5)

Body massive (figures 5*a* and 6*a,b,d*), translucent to pale yellowish in living specimens. Large, oval, black crystalline eyes. Cephalic appendages reduced; cirri interni and externi with reduced cirrophores merged with the flagellum, terminated with bi-, tri- and tetrafurcations (figures 14*c,d* and 18*b*). Cephalic papillae (secondary clavae) weakly outlined (figure 5*a*), imperceptible in SEM (figure 14*c,d*). *Cirri A* reduced and miniaturized, its cirrophore indistinctly merged with the rigid flagellum

**Table 3.** Measurements (in micrometres) of selected morphological structures of the adult females of *C. holmeni* (data pooled for populations from Atâ, Nuussuaq (Greenland), Igloolik (Turton Bay) and Kohke La Pass (Ladakh) mounted in polyvinyl-lactophenol. *N*, number of specimens/structures measured; range refers to the smallest and the largest structure among all measured specimens; s.d., standard deviation.

| character | N | range | | mean | | s.d. | |
|---|---|---|---|---|---|---|---|
| | | μm | *sp* | μm | *sp* | μm | *sp* |
| body length | 23 | 331–695 | *857–1143* | 519 | *1009* | 100 | *74* |
| scapular plate length | 23 | 30.3–63.9 | – | 51.7 | —— | 10.3 | —— |
| head appendages lengths | | | | | | | |
| cirrus internus | 22 | 7.8–16.0 | *16.7–28.1* | 12.1 | *23.7* | 2.3 | *2.6* |
| cephalic papilla | 23 | 7.5–12.2 | *16.9–26.9* | 10.5 | *20.8* | 1.2 | *2.7* |
| cirrus externus | 23 | 18.4–39.8 | *50.4–73.6* | 32.1 | *62.5* | 6.0 | *6.3* |
| clava | 14 | 5.3–10.7 | *12.0–22.6* | 8.4 | *15.8* | 1.6 | *2.7* |
| cirrus A | 23 | 19.9–45.2 | *56.6–77.1* | 35.2 | *68.3* | 7.1 | *5.3* |
| cirrus A/body length ratio | 23 | 6%–8% | – | 7% | —— | 1% | —— |
| body appendages lengths | | | | | | | |
| cirrus $C^l$ | 23 | 141.0–373.0 | *402.1–663.9* | 273.2 | *528.7* | 61.5 | *62.4* |
| cirrus $D^l$ | 21 | 130.0–412.0 | *368.7–719.0* | 258.4 | *493.1* | 76.8 | *91.6* |
| cirrus $D^d$ | 22 | 12.0–47.0 | *31.3–89.7* | 29.3 | *57.6* | 7.8 | *14.5* |
| spine ps | 23 | 8.0–29.0 | *14.4–58.4* | 16.8 | *33.6* | 5.1 | *11.2* |
| spine E | 22 | 17.0–56.0 | *47.3–97.7* | 36.9 | *71.6* | 9.9 | *12.5* |
| spine on leg I length | 22 | 3.9–11.8 | *10.7–20.8* | 8.5 | *16.7* | 2.1 | *2.7* |
| papilla on leg IV length | 22 | 5.0–9.4 | *11.6–19.9* | 7.7 | *15.1* | 1.4 | *1.9* |
| claw I heights | 23 | 19.0–38.1 | *51.8–77.6* | 31.6 | *61.9* | 5.2 | *6.7* |
| claw II heights | 23 | 17.9–41.7 | *51.7–77.6* | 31.7 | *62.0* | 5.4 | *6.8* |
| claw III heights | 23 | 17.4–41.0 | *52.7–73.9* | 31.9 | *62.4* | 5.3 | *6.1* |
| claw IV heights | 23 | 27.6–55.9 | *75.9–109.4* | 45.9 | *89.9* | 7.1 | *9.4* |

(figure 15a). Primary clavae miniaturized (figure 15a). Body appendage configuration A-C-D-$D^d$-E. All appendages but spine *E* in the form of setiform, short cirri. Lateral cirri usually longer than dorsal *cirri* $D^d$. Asymmetry in appendage development frequent in the positions C, D, $D^d$, the latter often doubled on one side of the body (figure 5a). Lateral *cirri* C and D often asymmetrically reduced to short, flexible filaments (figure 5a).

Sculpture covering evenly and densely the entire dorsal armour (figure 5a), comprising numerous and minute endocuticular pillars protruding as granules on the epicuticle (Figure 16a–c). Cephalic plate poorly developed, with central keel (figures 5a and 6a,b,d). Cervical (neck) plate visible in PCM as a dark grey belt preceding the scapular plate (figure 5a), devoid of pillars (figure 15a). Scapular plate reduced, narrow and with a weak W-suture (figures 5a and 16a). Pseudosegmental plate I′ narrower than the scapular plate, slightly broadening laterally (figures 5a, 6a,b,d and 16a). Median plate 1 bipartite, with both portions nearly equal in size (figures 5a, 6a,b and 16a), and flanked by the first pair of weakly sclerotized, supplementary lateral plates (figures 5a and 6d). First paired segmental plate tightly joined with the pseudosegmental plate II′, separated by epicuticular extension devoid of pillars (a smooth line in PCM; figures 5a, 6b and 16b). Median plate 2 bipartite; however, the epicuticular extension devoid of pillars in the centre of the anterior portion seems to divide this structure into three parts (figures 5a, 6b and 16b). The second pair of supplementary lateral plates flanking the most distal margins of median plate 2 (figures 5a and 6d). Second paired segmental plate and pseudosegmental plate III′ arranged similarly to the first pair on the dorsum (figures 5a, 6a,b,d and 16c). Median plate 3 undivided and rhomboid (figures 5a and 16c). Pseudosegmental

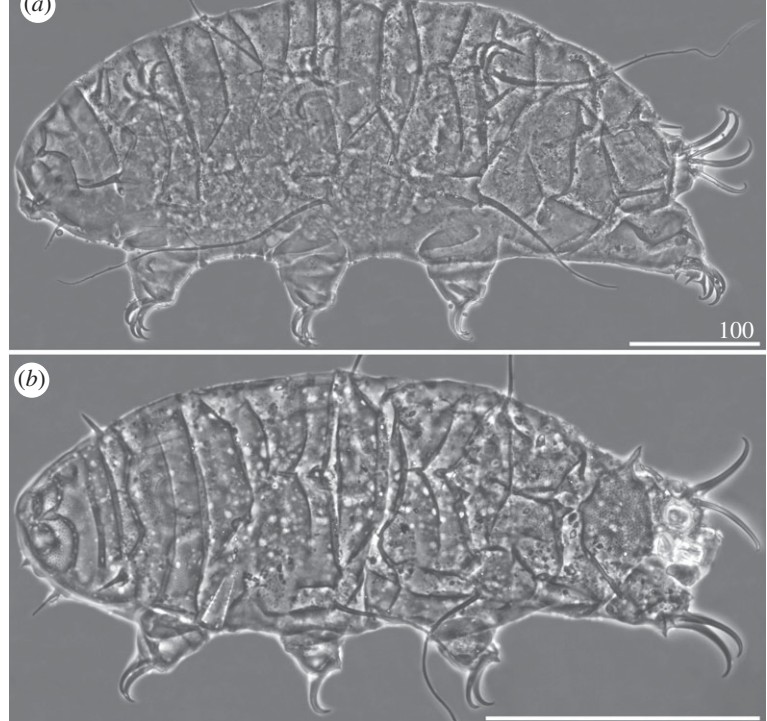

**Figure 4.** Habitus of *Cornechiniscus holmeni* (PCM): (*a*) female (dorsolateral view, specimen from Greenland); (*b*) larva (dorsolateral view, specimen from Greenland). All scale bars = 100 μm.

plate IV' undivided, narrow, with a continuous posterior margin (figures 5*a* and 6*a,b,d*). Caudal (terminal) plate with deep and strongly sclerotized incisions, continuous with base of spine E (figures 5*a* and 6*a,b,d*).

Venter smooth. Legs broad and short (figure 5*a*). Pedal (leg) plates I–IV present and densely encrusted with minute endocuticular pillars (figures 6*d* and 18*a*, pedal plates on legs IV visible only when the legs are extended). Spine I and spine constituting dentate collar IV short. Papilla on legs IV present (figure 5*a*). Claws strongly heteronych, spurless (figures 5*a* and 18*a–c*). Bases of external claws usually characteristically forked (figure 5*a*, legs III, IV). Gonopore sexpartite.

*Males* (i.e. from the third instar onwards; measurements and statistics in table 6)

Body smaller and more slender than in females (figures 5*b* and 6*c*). Spines I and IV often more robust and shorter than in females (figure 5*b*). Gonopore circular, with an arc-shaped slit (figure 21*a*).

*Juveniles* (i.e. the second instar, measurements and statistics in table 7)

Morphology similar to adults. A clear morphometric gap exists between juveniles and females, but there is no such gap between juveniles and males (compare tables 5–7).

*Larvae* (i.e. the first instar, measurements and statistics in table 8)

Morphology similar to adults, but plates do not cover the entire dorsum, and supplementary lateral plates are absent (figure 5*c*). Two-clawed larvae differ morphometrically from all subsequent instars (compare tables 5–8).

*Eggs*

Up to five eggs per exuvia were found.

*Molecular markers:* The species formed a separate monophyletic clade on the phylogenetic tree of the genus (figure 2). All three populations KG.012–14 exhibited a single ITS-1 haplotype (MT420865), but variability within COI was higher ($p$ = 0.7–2.0%), with three recovered haplotypes (KG.012: MT420451; KG.013: MT420452–4; KG.014: MT420455).

*Etymology:* From Latin *imperfectus* = unfinished. The name refers to the habitus of the new species, which resembles the only other appendaged congener, *C. holmeni*, but with much shorter lateral cirri and lacking spines on the posterior margin of the pseudosegmental plate, giving the appearance of an unfinished *C. holmeni*.

*Differential diagnosis:*

*Phenotypic*

**Table 4.** Measurements (in micrometres) of selected morphological structures of the larvae of *C. holmeni* from Atâ, Nuussuaq (Greenland) mounted in polyvinyl-lactophenol. *N*, number of specimens/structures measured; range refers to the smallest and the largest structure among all measured specimens; s.d., standard deviation.

| character | N | range | | mean | | s.d. | |
|---|---|---|---|---|---|---|---|
| | | µm | *sp* | µm | *sp* | µm | *sp* |
| body length | 2 | 252–257 | *889–1020* | 255 | *955* | 4 | *93* |
| scapular plate length | 2 | 24.7–28.9 | – | 26.8 | — | 3.0 | — |
| head appendages lengths | | | | | | | |
| cirrus internus | 2 | 6.5–6.7 | *23.2–26.3* | 6.6 | *24.7* | 0.1 | *2.2* |
| cephalic papilla | 2 | 6.5–8.2 | *26.3–28.4* | 7.4 | *27.3* | 1.2 | *1.5* |
| cirrus externus | 2 | 14.6–17.7 | *59.1–61.2* | 16.2 | *60.2* | 2.2 | *1.5* |
| clava | 2 | 5.1–6.4 | *20.6–22.1* | 5.8 | *21.4* | 0.9 | *1.1* |
| cirrus A | 2 | 12.0–14.4 | *48.6–49.8* | 13.2 | *49.2* | 1.7 | *0.9* |
| cirrus A/body length ratio | 2 | 5%–6% | — | 5% | — | 1% | — |
| body appendages lengths | 2 | 38%–45% | — | 41% | — | 5% | — |
| cirrus $C^l$ | | | | | | | |
| cirrus $D^l$ | 2 | 79.0–111.0 | *319.8–384.1* | 95.0 | *352.0* | 22.6 | *45.4* |
| spine $D^d$ | 2 | 87.0–110.0 | *352.2–380.6* | 98.5 | *366.4* | 16.3 | *20.1* |
| spine ps | 2 | 7.0–9.0 | *28.3–31.1* | 8.0 | *29.7* | 1.4 | *2.0* |
| spine E | 2 | 10.0–15.0 | *40.5–51.9* | 12.5 | *46.2* | 3.5 | *8.1* |
| spine on leg I length | 2 | 5.0–5.2 | *18.0–20.2* | 5.1 | *19.1* | 0.1 | *1.6* |
| papilla on leg IV length | 2 | 4.3–5.4 | *17.4–18.7* | 4.9 | *18.0* | 0.8 | *0.9* |
| claw I heights | 2 | 20.4–21.2 | *73.4–82.6* | 20.8 | *78.0* | 0.6 | *6.5* |
| claw II heights | 2 | 20.7–21.9 | *75.8–83.8* | 21.3 | *79.8* | 0.8 | *5.7* |
| claw III heights | 2 | 19.6–20.0 | *67.8–81.0* | 19.8 | *74.4* | 0.3 | *9.3* |
| claw IV heights | 2 | 32.4–33.1 | *114.5–131.2* | 32.8 | *122.9* | 0.5 | *11.8* |

Apart from the new species, there is only one other *Cornechiniscus* species with trunk appendages, *C. holmeni*, but females of the two species differ by a number of qualitative and quantitative traits:

— body appendage configuration: *A-C-D-D^d-E* in *C. imperfectus* sp. nov. versus *A-C-D-(D^d)-(ps)-E* in *C. holmeni*;

— the level of reduction of the horn-shaped *cirrus A*: 12.0–17.3 µm long (*29.7–43.3*), *cirrus A*/body length ratio 3–4% in *C. imperfectus* sp. nov. versus 19.9–45.2 µm long (*56.6–77.1*), *cirrus A*/body length ratio 6–8% in *C. holmeni*;

— relative lengths of peribuccal appendages: *cirri interni 27.9–39.0*, *cephalic papillae 26.8–37.3* and *cirri externi 40.0–50.8* in *C. imperfectus* sp. nov. versus *cirri interni 16.7–28.1*, *cephalic papillae 16.9–26.9* and *cirri externi 50.4–73.6* in *C. holmeni*;

— absolute and relative lengths of trunk appendages *C, D, D^d*: *cirrus C*: 38–86 µm (*91–206*), *cirrus D*: 48–72 µm (*131–191*), *cirrus D^d*: *84–132* in *C. imperfectus* sp. nov. versus *cirrus C*: 141–373 µm (*402–664*), *cirrus D*: 130–412 µm (*369–719*), *spine D^d*: *31–90* in *C. holmeni*.

Moreover, *C. imperfectus* sp. nov. is dioecious whereas in all *C. holmeni* populations reported so far only females were found, thus the species is most likely parthenogenetic.

Genotypic

The p-distances between *C. imperfectus* sp. nov. and other *Cornechiniscus* spp., for which genetic data are available, were as follows: ITS-1: from 7.7% (*C. lobatus*, MT420861–2) to 9.4% (*C. cornutus*, MT420858–60); COI: from 23.2% (*C. subcornutus*, MT420456–7) to 26.2% (*C. lobatus*, MT420440–9).

**Table 5.** Measurements (in micrometres) of selected morphological structures of the adult females of *C. imperfectus* sp. nov. mounted in Hoyer's medium. *N*, number of specimens/structures measured; range refers to the smallest and the largest structure among all measured specimens; s.d., standard deviation.

| character | *N* | range | | mean | | s.d. | | holotype | |
|---|---|---|---|---|---|---|---|---|---|
| | | µm | *sp* | µm | *sp* | µm | *sp* | µm | *sp* |
| body length | 15 | 420–518 | *1090–1249* | 467 | *1180* | 32 | *52* | 497 | *1243* |
| scapular plate length | 15 | 35.1–44.2 | – | 39.6 | — | 2.6 | — | 40.0 | — |
| head appendages lengths | | | | | | | | | |
| cirrus internus | 15 | 11.4–16.7 | *27.9–39.0* | 13.2 | *33.4* | 1.4 | *3.3* | 15.6 | *39.0* |
| cephalic papilla | 15 | 10.2–13.6 | *26.8–37.3* | 12.1 | *30.8* | 1.0 | *3.1* | 12.0 | *30.0* |
| cirrus externus | 14 | 15.1–18.9 | *40.0–50.8* | 17.2 | *43.5* | 1.0 | *3.3* | 17.2 | *43.0* |
| clava | 15 | 6.9–9.4 | *17.7–23.6* | 8.3 | *21.1* | 0.7 | *1.8* | 8.1 | *20.3* |
| cirrus A | 15 | 12.0–17.3 | *29.7–43.3* | 14.1 | *35.8* | 1.4 | *4.3* | 17.3 | *43.3* |
| cirrus A/body length ratio | 15 | 3%–4% | — | 3% | — | 0% | — | 3% | — |
| body appendages lengths | | | | | | | | | |
| cirrus $C^l$ | 15 | 38.3–85.9 | *91.2–205.5* | 60.5 | *153.3* | 12.9 | *32.2* | 52.0 | *130.0* |
| cirrus $D^l$ | 15 | 48.3–72.0 | *130.5–191.0* | 62.5 | *158.3* | 7.0 | *18.8* | 59.9 | *149.8* |
| cirrus $D^d$ | 15 | 35.0–52.9 | *83.8–132.4* | 43.7 | *110.8* | 6.0 | *17.0* | 47.3 | *118.3* |
| spine E | 14 | 11.9–34.4 | *28.3–95.2* | 21.6 | *54.7* | 8.3 | *20.9* | 25.3 | *63.3* |
| spine on leg I length | 15 | 5.0–6.9 | *12.0–18.5* | 6.0 | *15.3* | 0.5 | *1.6* | 5.7 | *14.3* |
| papilla on leg IV length | 15 | 5.5–7.7 | *13.8–20.5* | 6.6 | *16.7* | 0.6 | *2.0* | 6.8 | *17.0* |
| tooth on leg IV length | 14 | 3.7–7.1 | — | 5.4 | — | 1.1 | — | 7.1 | — |
| claw I heights | 15 | 25.1–31.0 | *66.3–74.7* | 27.8 | *70.2* | 1.8 | *2.9* | 28.5 | *71.3* |
| claw II heights | 15 | 24.5–32.5 | *63.8–78.3* | 27.5 | *69.5* | 2.3 | *4.2* | 28.7 | *71.8* |
| claw III heights | 15 | 25.0–33.9 | *65.2–76.7* | 28.0 | *70.7* | 2.6 | *4.0* | 29.6 | *74.0* |
| claw IV heights | 15 | 34.0–42.6 | *87.2–107.9* | 38.4 | *97.1* | 2.8 | *6.3* | 42.3 | *105.8* |

### 3.2.6. Cornechiniscus lobatus

*Cirri A* typical and well developed, with cirrophores thinner than the outer layer of the rigid *flagellum* (figure 7a). Occasional, minute spicules in lateral positions *C–E* (figure 18d). Pseudosegmental plates I–III' indistinctly merged with scapular and paired segmental plates (figures 7a and 8). Median plates 1–2 bipartite, with clearly larger and strongly sculptured anterior portions, whereas median plate 3 unipartite, rhomboidal and well developed (figures 7a and 8a,b). Posterior margin of the second paired segmental plate broad (figure 7a). The paired pseudosegmental plate IV' usually with a strongly sclerotized posterior margin bearing at least one lobe [30], either semicircular or terminated with 1–2 apices (figures 7a and 8a). Sculpture consisting of endocuticular pillars most discernible on the central part of the dorsum (figures 7a and 8a). Supplementary lateral plates undistinguishable in PCM (figure 7a), but identifiable in SEM (figure 8b,c). Venter with evident and deep cuticular grooves (figure 8d). Spine I and spine constituting dentate collar IV short and bluntly terminated (figure 7a). Pedal (leg) plates usually non-identifiable even on legs IV. Claws slightly heteronych (anisonych), spurless.

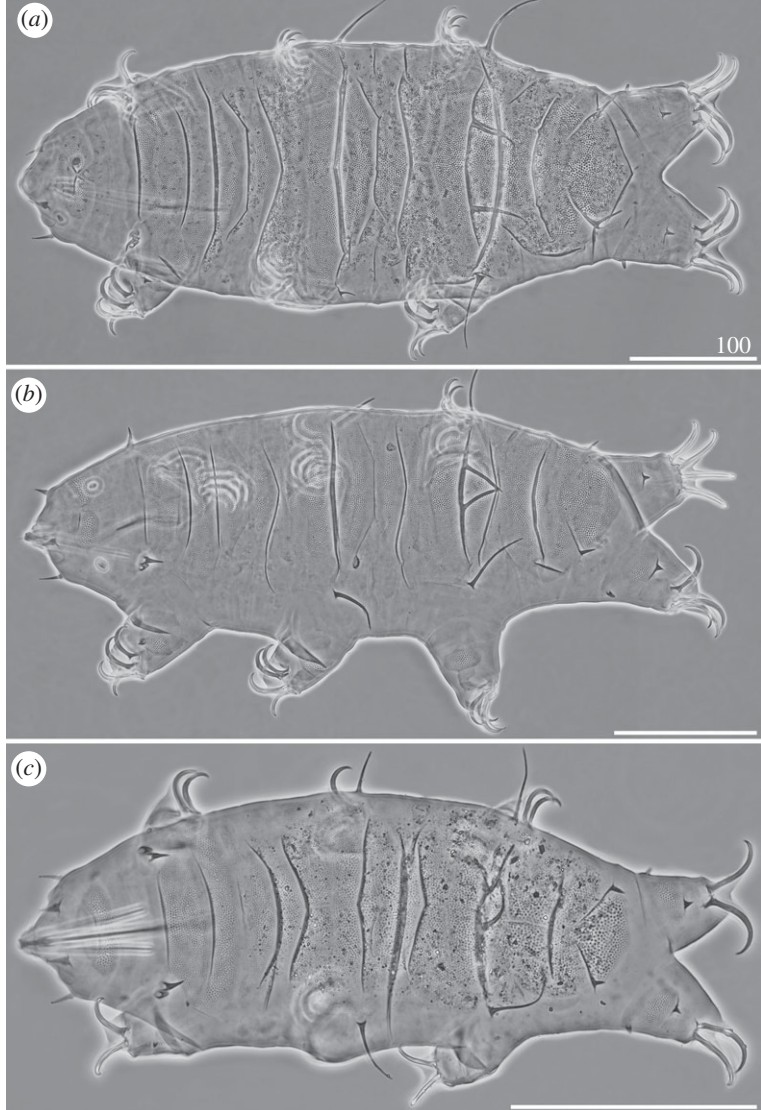

**Figure 5.** Habitus of *Cornechiniscus imperfectus* sp. nov. (PCM): (*a*) female (holotype, dorsal view); (*b*) male (allotype, dorsolateral view); (*c*) larva (paratype, dorsal view). All scale bars = 100 μm.

Larvae with already developed adult morphology (figure 7b), except for the lack of the gonopore and anus. Pseudosegmental posterior lobe usually terminated with one apex. Clear sutures (cuticular extensions) separating posteriormost and lateralmost portions of the scapular plate in both adults and larvae (figures 7 and 8b,c).

### 3.2.7. Cornechiniscus madagascariensis

*Cirri A* typical, with well-delimited, lighter cirrophores (figures 9a,b and 10a). Margins of all dorsal plates strongly sclerotized in adult females (figures 9a,b and 10a), but indistinct in juveniles (figure 9c). The W-shaped suture on the scapular plate indiscernible (figure 9a–c). Median plates 1–2 bipartite, with dominant anterior portions and strongly reduced posterior portions; median plate 3 large and well developed (figure 9a,b). Posterior margin of the second paired segmental plate broad (figure 9b) and may form a lobe (figure 9c). Pseudosegmental plate IV' bipartite and with a broadened posterior edge forming a lobe, terminated with 1–2 apices (figure 9a,b). Especially in juveniles, the pseudosegmental lobe may be absent (figure 9c). Caudal incisions poorly marked and with occasional spicules in position E (figures 9a,b and 16f). Single supplementary lateral plates flanking median plates 1–2 present and strongly sclerified (figures 9a,b and 10a). Sculpture evident and autapomorphic, consisting of large, sparsely distributed endocuticular pillars connected by *striae* (figures 9a,c and 10a) [31]. Venter with deep cuticular grooves (figure 10b). Spine I present. Only pedal plates IV developed, with

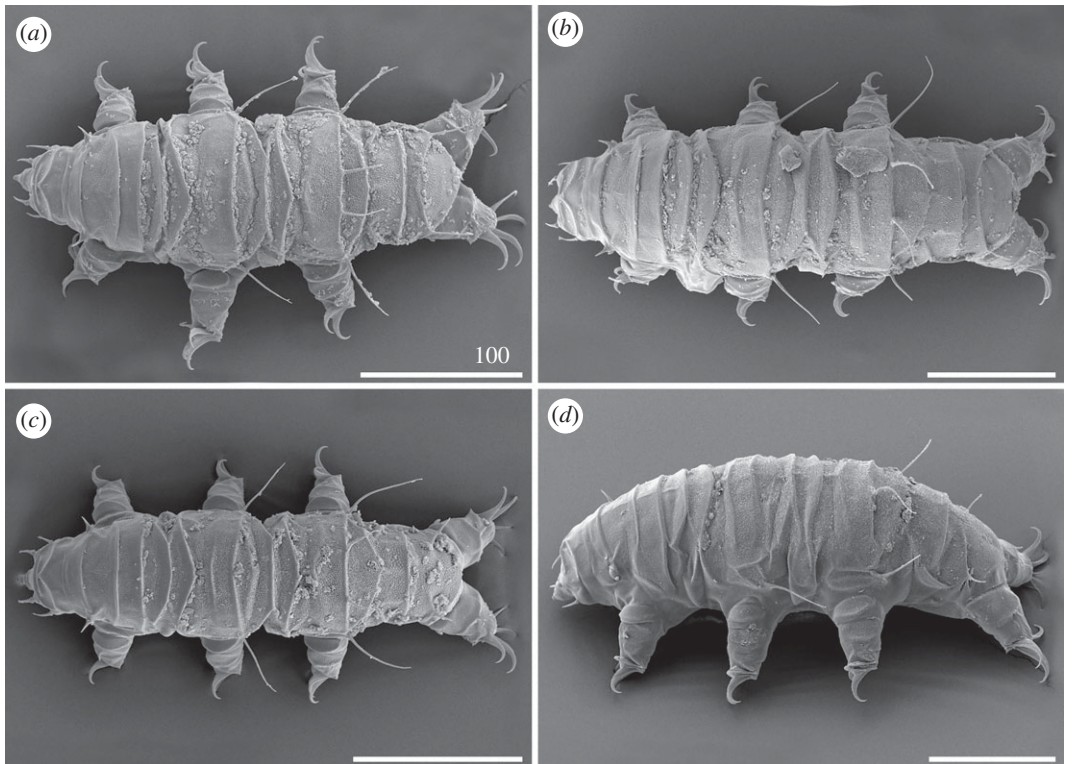

**Figure 6.** Habitus of *Cornechiniscus imperfectus* sp. nov. (SEM): (*a,b*) females (paratypes, dorsal view); (*c*) male (paratype, dorsal view); (*d*) female (paratype, lateral view). All scale bars = 100 µm.

finer pillars adjacent to the single tooth constituting dentate collar IV (figure 9*a,c*). Claws slightly heteronych (anisonych), spurless.

Larvae very similar to those of *C. lobatus* (compare figures 7*b* and 9*d*) since the striation is not developed yet in this life stage. However, the pillars in central portions of dorsal plates are larger.

### 3.2.8. Cornechiniscus schrammi

*Cirri A* elongated and slender, with well-delimited, thinner and lighter cirrophores (figure 11). Dorsum with strongly sclerotized plates, and both pseudosegmental plates I–III′ and supplementary lateral plates well delimited from the other structures. Sculpture minute and faint, comprising tightly arranged, polygonal endocuticular pillars [32]. Venter with deep cuticular grooves. Spines I reduced and minute, dentate collar IV absent. Claws slightly heteronych (anisonych), spurless.

Sexual dimorphism limited to a different gonopore shape (circular in males and sexpartite in females) and to a smaller body size (length and width) in males than females (compare figure 11*a,b*) [32].

### 3.2.9. Cornechiniscus subcornutus

*Cirri A* strongly reduced, with indistinct cirrophores (figure 12). Margins of all dorsal plates well marked. The W-shaped suture closely positioned to the anterior edge of the scapular plate. Median plate 1 bipartite, median plate 2 tripartite and median plate 3 unipartite (figures 12 and 13). Pseudosegmental plates II–III′ indistinctly merged with paired segmental plates. Pseudosegmental plate IV′ bipartite and with two short, blunt teeth (figure 12*a–c*, arrowheads). In juveniles, pseudosegmental teeth absent (figure 12*d*). Caudal incisions strongly marked (figures 12*a–c* and 13), and meeting in the central portion of the caudal (terminal) plate, almost dividing it in two parts (figure 12*c*). Single supplementary lateral plates flanking median plates 1–2 strongly sclerified (figures 12*a–c* and 13). Sculpture evident and consisting of large endocuticular pillars [25]. Venter with deep cuticular grooves. Only pedal plates IV developed, with finer pillars adjacent to the single tooth constituting dentate collar IV (figure 12). Spine I reduced in adults (figures 12*c* and 13*b*), absent in juveniles (figure 12*d*). Claws extremely elongated (tables 9–11) and strongly heteronych, external claws usually bear long spurs directed upwards.

**Table 6.** Measurements (in micrometres) of selected morphological structures of the adult males of *C. imperfectus* sp. nov. mounted in Hoyer's medium. *N*, number of specimens/structures measured; range refers to the smallest and the largest structure among all measured specimens; s.d., standard deviation.

| character | N | range | | mean | | s.d. | | allotype | |
|---|---|---|---|---|---|---|---|---|---|
| | | μm | *sp* | μm | *sp* | μm | *sp* | μm | *sp* |
| body length | 15 | 284–473 | *1114–1318* | 347 | *1187* | 57 | *67* | 413 | *1255* |
| scapular plate length | 15 | 23.6–36.8 | — | 29.2 | — | 4.5 | — | 32.9 | — |
| head appendages lengths | | | | | | | | | |
| cirrus internus | 15 | 7.6–13.8 | *25.8–39.0* | 10.1 | *34.8* | 1.7 | *3.7* | 8.5 | *25.8* |
| cephalic papilla | 15 | 7.7–12.9 | *30.2–39.9* | 9.9 | *33.8* | 1.6 | *2.9* | 12.0 | *36.5* |
| cirrus externus | 15 | 10.1–16.8 | *42.4–49.4* | 13.2 | *45.2* | 2.0 | *2.2* | 14.2 | *43.2* |
| clava | 15 | 4.6–9.0 | *19.3–24.9* | 6.6 | *22.7* | 1.2 | *1.6* | 7.1 | *21.6* |
| cirrus A | 15 | 7.6–13.4 | *31.9–40.6* | 10.4 | *35.7* | 1.7 | *2.3* | 11.9 | *36.2* |
| cirrus A/body length ratio | 15 | 2%–3% | — | 3% | — | 0% | — | 3% | — |
| body appendages lengths | | | | | | | | | |
| cirrus $C^l$ | 15 | 29.5–63.0 | *103.0–203.2* | 42.1 | *143.3* | 11.1 | *27.1* | 33.9 | *103.0* |
| cirrus $D^l$ | 15 | 31.9–72.2 | *122.1–213.4* | 47.3 | *160.4* | 13.6 | *29.8* | 49.6 | *150.8* |
| cirrus $D^d$ | 15 | 29.2–56.2 | *94.2–172.9* | 39.8 | *137.8* | 6.8 | *23.2* | 32.1 | *97.6* |
| spine E | 12 | 7.5–26.9 | *26.2–77.2* | 14.7 | *49.4* | 6.7 | *20.0* | 11.1 | *33.7* |
| spine on leg I length | 15 | 2.6–7.6 | *10.4–21.2* | 4.5 | *15.2* | 1.3 | *2.8* | 6.0 | *18.2* |
| papilla on leg IV length | 15 | 4.0–6.9 | *15.2–22.9* | 5.4 | *18.5* | 0.7 | *2.2* | 6.9 | *21.0* |
| tooth on leg IV length | 13 | 3.4–7.4 | — | 5.1 | — | 1.1 | — | 5.1 | — |
| claw I heights | 15 | 18.7–28.0 | *69.7–85.2* | 22.1 | *76.0* | 3.0 | *4.2* | 26.2 | *79.6* |
| claw II heights | 15 | 17.5–26.9 | *66.3–82.2* | 21.5 | *74.0* | 2.9 | *4.4* | 23.8 | *72.3* |
| claw III heights | 15 | 17.7–27.9 | *67.0–79.0* | 21.6 | *74.0* | 3.5 | *3.4* | 25.4 | *77.2* |
| claw IV heights | 15 | 23.5–38.4 | *89.0–122.5* | 30.9 | *106.2* | 4.4 | *8.5* | 36.4 | *110.6* |

### 3.2.10. *Cornechiniscus tibetanus*

*Cirri A* reduced, with indistinguishable cirrophores. Two pairs of supplementary lateral plates on each side of the median plates 1–2. Sculpture evident and consisting of large endocuticular pillars protruding through epicuticle as granules of similar size on all plates [33]. Venter with deep cuticular grooves. Dentate collar on legs IV reduced, with two short teeth [33]. Claws strongly heteronych, massive and robust; spurless.

## 3.3. Comparative morphology and anatomy of *Cornechiniscus* species

### 3.3.1 Cephalic appendages

Cephalic papillae (secondary clavae) are rarely well marked and large (figure 14*a*), they are usually poorly delimited from the cephalic cuticle (figure 14*b*,*e*–*f*), and sometimes indiscernible (figure 14*c*,*d*). Mouth cone divided into three segments (figure 14*b*–*e*). All peribuccal cirri with reduced cirrophores indistinctly merged with the flagellum, which can be either bi-, trifurcated or tufted at its tip. Cirri interni either bulbous (figure 14*a*,*b*, *e*–*f*) or conical (figure 14*c*,*d*). Cirri externi elongated (figure 14).

**Table 7.** Measurements (in micrometres) of selected morphological structures of the juveniles of *C. imperfectus* sp. nov. mounted in Hoyer's medium. *N*, number of specimens/structures measured; range refers to the smallest and the largest structure among all measured specimens; s.d., standard deviation.

| | | range | | mean | | s.d. | |
|---|---|---|---|---|---|---|---|
| character | N | μm | sp | μm | sp | μm | sp |
| body length | 5 | 341–381 | 1129–1172 | 366 | 1153 | 18 | 18 |
| scapular plate length | 5 | 30.2–33.1 | — | 31.8 | — | 1.5 | — |
| head appendages lengths | | | | | | | |
| cirrus internus | 5 | 9.9–11.7 | 30.4–36.8 | 10.9 | 34.2 | 0.7 | 2.9 |
| cephalic papilla | 5 | 10.0–12.0 | 30.5–39.7 | 10.8 | 34.1 | 0.8 | 3.6 |
| cirrus externus | 5 | 13.5–14.8 | 42.3–45.4 | 14.1 | 44.3 | 0.6 | 1.2 |
| clava | 5 | 6.5–7.3 | 21.3–23.5 | 7.0 | 22.0 | 0.3 | 0.9 |
| cirrus A | 5 | 9.6–12.3 | 29.4–37.5 | 11.1 | 35.0 | 1.0 | 3.3 |
| cirrus A/body length ratio | 5 | 3%–3% | — | 3% | — | 0% | — |
| body appendages lengths | | | | | | | |
| cirrus $C^l$ | 4 | 48.8–61.3 | 161.6–186.9 | 54.5 | 169.2 | 5.1 | 11.9 |
| cirrus $D^l$ | 4 | 48.7–64.0 | 161.3–193.4 | 56.9 | 179.4 | 8.0 | 16.4 |
| cirrus $D^d$ | 5 | 37.0–55.9 | 122.5–170.4 | 46.2 | 145.1 | 6.9 | 17.5 |
| spine E | 5 | 5.8–26.4 | 19.2–81.0 | 17.8 | 55.8 | 9.0 | 27.5 |
| spine on leg I length | 5 | 3.1–4.8 | 10.3–14.9 | 4.3 | 13.4 | 0.7 | 1.9 |
| papilla on leg IV length | 5 | 3.6–6.4 | 11.9–19.5 | 5.6 | 17.4 | 1.1 | 3.2 |
| tooth on leg IV length | 4 | 4.7–6.6 | — | 5.5 | — | 1.0 | — |
| claw I heights | 5 | 19.8–25.3 | 65.6–76.4 | 22.3 | 70.0 | 2.0 | 4.2 |
| claw II heights | 4 | 21.0–24.2 | 66.9–73.1 | 22.5 | 70.0 | 1.4 | 2.6 |
| claw III heights | 4 | 21.4–24.3 | 65.5–73.4 | 22.3 | 69.2 | 1.4 | 4.2 |
| claw IV heights | 4 | 30.9–35.7 | 97.5–107.9 | 32.8 | 101.9 | 2.1 | 4.4 |

Primary clavae reduced, either adjacent to *cirri A* (figure 15a,c,d) or merged with the cirrophore base (figure 15b). Two types of horn-shaped *cirri A* present: typical, with a clear division between the cirrophore (either smooth or sculptured) and the rigid flagellum (figure 15b,c), or miniaturized, with the cirrophore and flagellum forming a continuous appendage (figure 15a,d).

### 3.3.2. Sculpture of dorsal armour

Dorsal plate sculpturing is formed by endocuticular pillars protruding through epicuticle as granules, which may differ in size and distribution: they can either be small and densely arranged (figure 16a–c), or large and more sparsely distributed (figure 16d–e). Two autapomorphic subtypes can be distinguished: (i) large, roundish pillars joined by *striae* in *C. madagascariensis* (figure 16f), with each pillar connected by 3–6 *striae* with the neighbouring pillars (figure 16g); (ii) large to very large, almost ideally round and clearly separated pillars (manifested as granules on the cuticle surface) without *striae* in *C. ceratophorus* (see [28]). Typically for the *Pseudechiniscus*-like genera, epicuticle is particularly thin and prone to being torn off and exposing the endocuticle when subjected to SEM preparation (figure 16g–h).

### 3.3.3. Claw morphology

Claws always have elongated branches only mildly bent in their distal portions (figures 17–19). There are three categories of proportion of length of claws I–III and claws IV: isonych, with no differences in claw lengths (present only in *C. cornutus*; figure 17c,d); slightly heteronych (anisonych), with claws IV little

**Table 8.** Measurements (in micrometres) of selected morphological structures of the larvae of *C. imperfectus* sp. nov. mounted in Hoyer's medium. *N*, number of specimens/structures measured; range refers to the smallest and the largest structure among all measured specimens; s.d., standard deviation.

| character | *N* | range μm | range *sp* | mean μm | mean *sp* | s.d. μm | s.d. *sp* |
|---|---|---|---|---|---|---|---|
| body length | 3 | 212–261 | *1145–1286* | 245 | *1221* | 28 | *71* |
| scapular plate length | 3 | 17.2–22.8 | — | 20.1 | — | 2.8 | — |
| head appendages lengths | | | | | | | |
| cirrus internus | 3 | 6.0–8.9 | *34.9–40.4* | 7.7 | *38.1* | 1.5 | *2.9* |
| cephalic papilla | 3 | 5.5–8.0 | *31.1–39.4* | 6.9 | *34.2* | 1.3 | *4.6* |
| cirrus externus | 3 | 7.0–11.9 | *40.7–52.2* | 9.7 | *47.7* | 2.5 | *6.2* |
| clava | 2 | 4.7–5.6 | *20.6–27.6* | 5.2 | *24.1* | 0.6 | *4.9* |
| cirrus *A* | 3 | 5.8–8.0 | *33.7–35.1* | 6.9 | *34.3* | 1.1 | *0.7* |
| cirrus *A*/body length ratio | 3 | 3%–3% | — | 3% | — | 0% | — |
| body appendages lengths | | | | | | | |
| cirrus *C$^l$* | 3 | 24.0–30.3 | *132.9–139.5* | 27.4 | *136.6* | 3.2 | *3.4* |
| cirrus *D$^l$* | 3 | 24.0–31.3 | *137.3–152.7* | 28.8 | *143.2* | 4.1 | *8.3* |
| cirrus *D$^d$* | 3 | 22.1–31.0 | *124.1–136.0* | 26.1 | *129.5* | 4.5 | *6.0* |
| spine *E* | 2 | 5.6–8.4 | *27.6–36.8* | 7.0 | *32.2* | 2.0 | *6.5* |
| spine on leg I length | 3 | 2.1–2.6 | *10.3–12.2* | 2.3 | *11.3* | 0.3 | *0.9* |
| papilla on leg IV length | 3 | 3.5–3.9 | *15.4–22.1* | 3.7 | *18.9* | 0.2 | *3.4* |
| tooth on leg IV length | 3 | 2.2–6.6 | — | 4.1 | — | 2.3 | — |
| claw I heights | 3 | 15.8–16.8 | *69.3–91.9* | 16.1 | *81.3* | 0.6 | *11.4* |
| claw II heights | 3 | 14.8–16.6 | *64.9–86.6* | 15.4 | *77.8* | 1.0 | *11.4* |
| claw III heights | 3 | 14.4–15.9 | *69.7–83.7* | 15.3 | *76.9* | 0.8 | *7.0* |
| claw IV heights | 3 | 21.7–25.1 | *97.4–126.2* | 23.0 | *115.7* | 1.8 | *15.9* |

longer than claws I–III (the most frequent type: *C. brachycornutus*, *C. ceratophorus*, *C. lobatus*, *C. madagascariensis* and *C. schrammi*; figures 17*a*,*b*, 18*d*–*f* and 19*a*,*b*); and strongly heteronych, with claws IV much longer than claws I–III (*C. holmeni*, *C. imperfectus* sp. nov., *C. subcornutus*, *C. tibetanus*; figures 17*e*,*f*, 18*a*–*c* and 19*c*–*f*). In parallel to claw proportions, two species have an autapomorphic claw morphology: *C. cornutus*, with all claws miniaturized and delicate (figure 17*c*,*d*); and *C. tibetanus*, with particularly robust and massive claws on all legs (figure 19*f*).

Claw spurs are rarely present in *Cornechiniscus*. The only known species with symmetrical spurs pointing upwards on external claws IV is *C. subcornutus* (figure 19*d*). In other species, spurs are either always absent (*C. cornutus*, *C. imperfectus* sp. nov., *C. lobatus*, *C. madagascariensis*, *C. schrammi*, *C. tibetanus*) or are aberrations in the form of small spurs on only one of the external claws IV, and thus far they have been recorded exclusively in *C. brachycornutus*, *C. ceratophorus* and *C. holmeni* (in the latter, in contrast to the original description, single spurs are sometimes also found on internal claws).

### 3.3.4. Bucco-pharyngeal apparatus

The buccal apparatus is more delicate and with a thinner buccal tube in smaller species (e.g. *C. lobatus*; figure 20*a*) and proportionally larger and with a broader buccal tube in large representative of the genus (e.g. *C. imperfectus* sp. nov.; figure 20*b*). The buccal ring surrounds the mouth opening, the buccal crown is conical (figure 20*c*). Stylets with longitudinal grooves (figure 20*a*,*b*), and supports encrusted with $CaCO_3$ that quickly dissolve during the sodium hypochlorite extraction for SEM (figure 20*a*). Buccal tube of uniform thickness from the posterior portion of the buccal crown to slightly before the level of

**Table 9.** Measurements (in micrometres) of selected morphological structures of the adult females of *C. subcornutus* from Kyrgyzstan, mounted in Hoyer's medium. *N*, number of specimens/structures measured; range refers to the smallest and the largest structure among all measured specimens; s.d., standard deviation.

| character | N | range | | mean | | s.d. | |
|---|---|---|---|---|---|---|---|
| | | µm | *sp* | µm | *sp* | µm | *sp* |
| body length | 16 | 252–386 | *632–763* | 330 | *691* | 38 | *40* |
| scapular plate length | 16 | 39.9–56.1 | — | 47.9 | — | 5.5 | — |
| head appendages lengths | | | | | | | |
| cirrus internus | 16 | 5.3–8.2 | *10.5–19.5* | 6.8 | *14.3* | 0.9 | *2.4* |
| cephalic papilla | 16 | 7.1–9.2 | *13.9–22.4* | 8.0 | *16.9* | 0.6 | *2.1* |
| cirrus externus | 16 | 13.9–18.7 | *30.3–40.0* | 16.8 | *35.3* | 1.3 | *2.9* |
| clava | 11 | 7.3–10.3 | *13.0–20.3* | 8.5 | *17.0* | 0.9 | *2.3* |
| cirrus A | 16 | 11.9–17.9 | *24.5–36.4* | 14.4 | *30.1* | 1.8 | *2.9* |
| cirrus A/body length ratio | 16 | 3%–6% | — | 4% | — | 1% | — |
| body appendages lengths | | | | | | | |
| spine on leg I length | 10 | 3.0–7.3 | *6.1–14.4* | 5.0 | *10.0* | 1.2 | *2.5* |
| papilla on leg IV length | 14 | 4.6–7.6 | *9.2–15.0* | 5.9 | *12.2* | 0.9 | *1.8* |
| claw I heights | 15 | 17.5–22.1 | *36.8–51.5* | 19.9 | *41.5* | 1.6 | *4.0* |
| claw II heights | 15 | 17.3–22.7 | *35.6–48.8* | 20.0 | *41.5* | 1.6 | *4.0* |
| claw III heights | 15 | 17.2–22.0 | *36.6–53.7* | 19.9 | *41.5* | 1.6 | *4.5* |
| claw IV heights | 16 | 25.9–37.9 | *58.2–92.4* | 32.4 | *68.1* | 3.3 | *8.1* |

**Table 10.** Measurements (in micrometres) of selected morphological structures of the adult males of *C. subcornutus* from Kyrgyzstan, mounted in Hoyer's medium. *N*, number of specimens/structures measured; range refers to the smallest and the largest structure among all measured specimens; s.d., standard deviation.

| character | N | range | | mean | | s.d. | |
|---|---|---|---|---|---|---|---|
| | | µm | *sp* | µm | *sp* | µm | *sp* |
| body length | 3 | 281–317 | *704–757* | 297 | *722* | 18 | *30* |
| scapular plate length | 3 | 39.8–41.9 | — | 41.1 | — | 1.1 | — |
| head appendages lengths | | | | | | | |
| cirrus internus | 3 | 5.4–6.7 | *13.6–16.0* | 6.1 | *14.8* | 0.7 | *1.2* |
| cephalic papilla | 3 | 8.3–9.4 | *20.0–22.4* | 8.7 | *21.1* | 0.6 | *1.3* |
| cirrus externus | 3 | 13.3–15.0 | *32.0–35.8* | 14.1 | *34.4* | 0.9 | *2.1* |
| clava | 3 | 6.9–8.0 | *16.5–19.2* | 7.4 | *17.9* | 0.6 | *1.4* |
| cirrus A | 3 | 10.8–12.8 | *27.1–30.8* | 12.1 | *29.3* | 1.1 | *1.9* |
| cirrus A/body length ratio | 3 | 4%–4% | – | 4% | — | 0% | — |
| body appendages lengths | | | | | | | |
| spine on leg I length | 3 | 4.4–5.5 | *10.5–13.2* | 4.8 | *11.8* | 0.6 | *1.4* |
| papilla on leg IV length | 3 | 4.9–6.3 | *12.3–15.0* | 5.7 | *13.8* | 0.7 | *1.4* |
| claw I heights | 2 | 18.7–22.0 | *47.0–52.5* | 20.4 | *49.7* | 2.3 | *3.9* |
| claw II heights | 3 | 19.5–21.6 | *49.0–51.9* | 20.7 | *50.3* | 1.1 | *1.5* |
| claw III heights | 3 | 19.0–21.7 | *47.7–51.8* | 20.5 | *49.8* | 1.4 | *2.0* |
| claw IV heights | 3 | 33.9–39.7 | *84.2–95.4* | 36.3 | *88.3* | 3.0 | *6.2* |

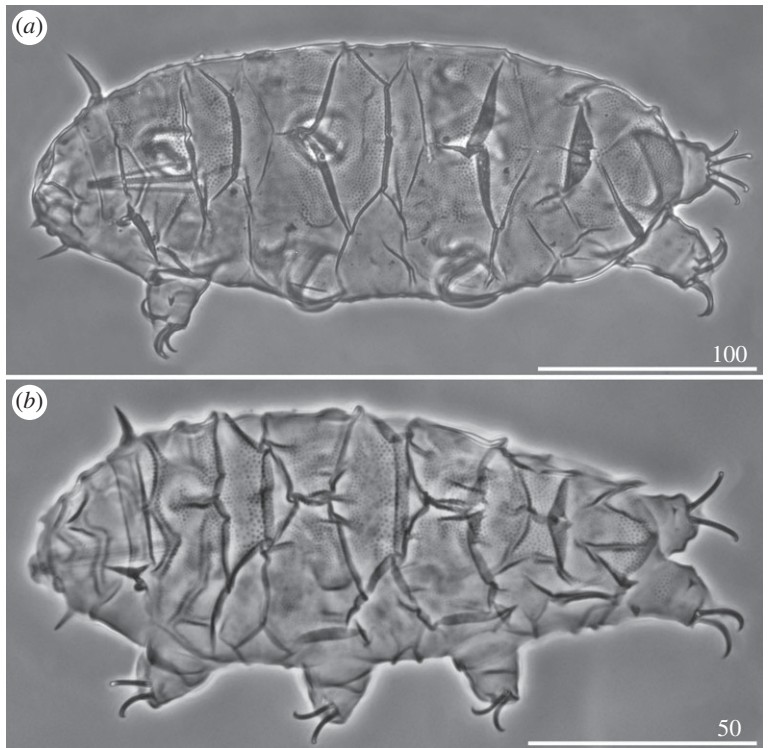

**Figure 7.** Habitus of *Cornechiniscus lobatus* (PCM): (*a*) female (dorsolateral view, specimen from France); (*b*) larva (dorsolateral view, specimen from France). All scale bars in μm.

**Table 11.** Measurements (in micrometres) of selected morphological structures of the adult males of *C. subcornutus* from Kyrgyzstan, mounted in Hoyer's medium. *N*, number of specimens/structures measured; range refers to the smallest and the largest structure among all measured specimens; s.d., standard deviation.

| character | *N* | range | | mean | | s.d. | |
|---|---|---|---|---|---|---|---|
| | | μm | *sp* | μm | *sp* | μm | *sp* |
| body length | 4 | 213–236 | 650–727 | 226 | 689 | 10 | 33 |
| scapular plate length | 4 | 29.3–36.3 | — | 33.0 | — | 3.0 | — |
| head appendages lengths | | | | | | | |
| cirrus internus | 4 | 3.3–4.6 | 10.3–13.7 | 4.0 | 12.1 | 0.5 | 1.4 |
| cephalic papilla | 4 | 5.6–6.5 | 16.8–19.1 | 6.0 | 18.3 | 0.4 | 1.1 |
| cirrus externus | 4 | 9.8–10.1 | 27.0–34.1 | 10.0 | 30.5 | 0.1 | 3.1 |
| clava | 3 | 3.5–5.9 | 11.9–16.3 | 4.8 | 14.2 | 1.2 | 2.2 |
| cirrus *A* | 4 | 7.4–10.1 | 25.3–30.0 | 9.2 | 27.8 | 1.2 | 2.3 |
| cirrus *A*/body length ratio | 4 | 3%–4% | — | 4% | — | 0% | — |
| body appendages lengths | | | | | | | |
| spine on leg I length | 3 | 2.5–3.8 | 8.5–11.9 | 3.3 | 10.2 | 0.7 | 1.7 |
| papilla on leg IV length | 4 | 3.1–4.6 | 10.6–13.1 | 4.0 | 12.2 | 0.6 | 1.1 |
| claw I heights | 4 | 13.1–16.4 | 40.5–48.0 | 14.4 | 43.6 | 1.6 | 3.4 |
| claw II heights | 4 | 12.1–15.8 | 37.8–46.2 | 13.6 | 41.3 | 1.8 | 3.6 |
| claw III heights | 4 | 10.9–15.4 | 34.1–45.0 | 13.0 | 39.3 | 2.2 | 4.5 |
| claw IV heights | 3 | 18.6–23.1 | 59.7–63.6 | 20.3 | 62.3 | 2.5 | 2.2 |

condyles of furcae, where the longitudinal dorsal crest begins. Then, on the lateral sides, the buccal tube is flanked by two lobe-like crests (figure 20*a,b*). The posteriormost portion of the buccal tube is tapered and flexible (figure 20*a,b*). The pharynx is fusiform, with reduced placoids in a form of jagged ridges (figure 20*d*).

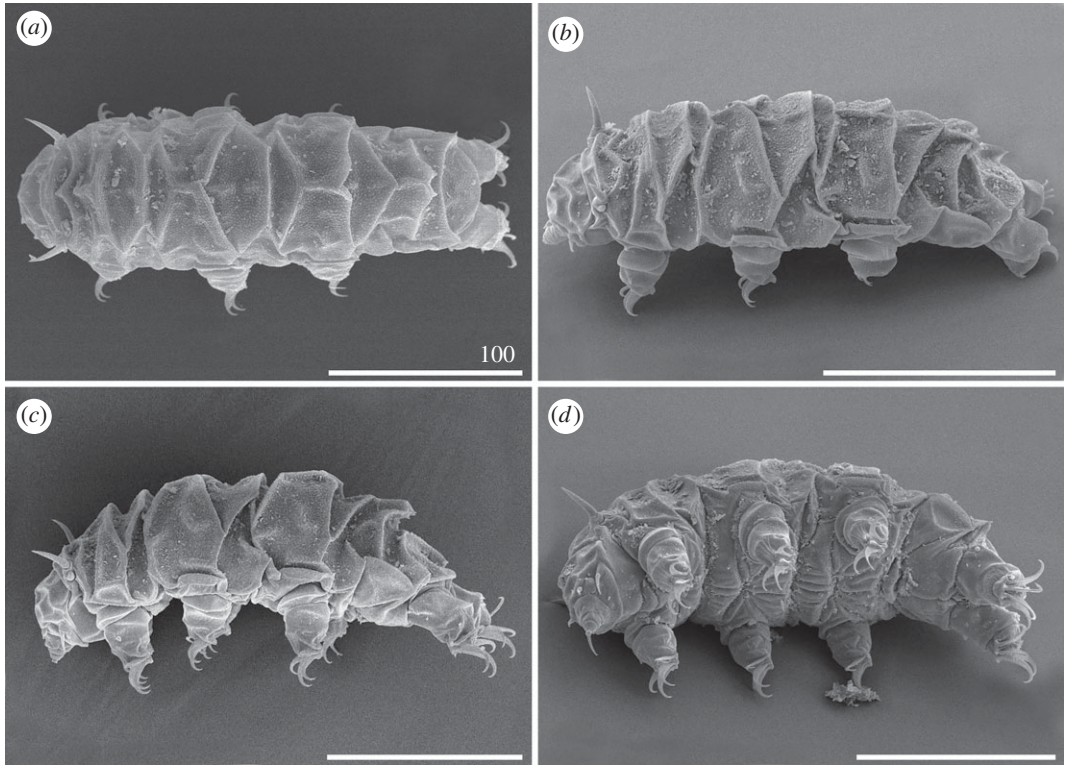

**21**

**Figure 8.** Habitus of *Cornechiniscus lobatus* (females from Kyrgyzstan, SEM): (*a*) dorsal view; (*b*) dorsolateral view; (*c*) lateral view; (*d*) ventral view. All scale bars = 100 µm.

### 3.3.5. Sexual dimorphism in dioecious (gonochoristic) species

Only 3 out of the 10 described species of *Cornechiniscus* are known to be dioecious: *C. imperfectus* sp. nov., *C. schrammi* and *C. subcornutus*. In contrast to echiniscid genera with exclusively dioecious species, such as *Hypechiniscus* or *Antechiniscus*, sexual dimorphism in *Cornechiniscus* is barely recognizable. The primary difference is the shape of the gonophore, which in females is sexpartite and in the male is round, with a semicircular slit; differing from the circular gonoporal opening that is typical for the majority of echiniscid genera (figure 21; see also Rebecchi *et al.* [15] for another example of a semicircular slit observed in *Novechiniscus*). Males are also slightly shorter only in *C. imperfectus* sp. nov. (tables 5 and 6) and in all three species more slender than females (figures 5*a,b*, 6, 11 and 12*a–c*).

## 4. Discussion

### 4.1. Phylogeny of Echiniscidae versus classification proposed in Guil *et al.* [24]

The history of how echiniscid genera were classified and distinguished dates back to the fundamental works of Thulin [7,8] who differentiated *Pseudechiniscus* from *Echiniscus* using the presence of the pseudosegmental plate IV' in *Pseudechiniscus*, and distinguished *Bryodelphax* from *Echiniscus* by the absence of caudal incisions and by a different division of median plates in *Bryodelphax*. However, no formal hypothesis on the relationships within Echiniscidae was posed until the crucial monograph by Kristensen [22]. In that work, the terms *Echiniscus*-line (i.e. *Bryodelphax*, *Echiniscus*, *Testechiniscus*) and *Pseudechiniscus*-line (i.e. *Antechiniscus*, *Cornechiniscus*, *Mopsechiniscus*, *Proechiniscus*, *Pseudechiniscus*) were coined, and the genera *Parechiniscus* and *Novechiniscus*, with poorly differentiated dorsal armour, were inferred as the basal taxa. Kristensen followed the traditional usage of the dorsal plate arrangement and diversification in the erection of new genera but, importantly, he also initiated the application of new taxonomic criteria into the echiniscid classification at the supra-specific level, such as sense organs (especially eye colour and structure), morphology of the bucco-pharyngeal apparatus, and ventral armature. Later, Jørgensen [23] revisited the phylogeny of the family by formal cladistic testing, which suggested two, surprising at that time, changes in the cladogram: *Bryodelphax* outside

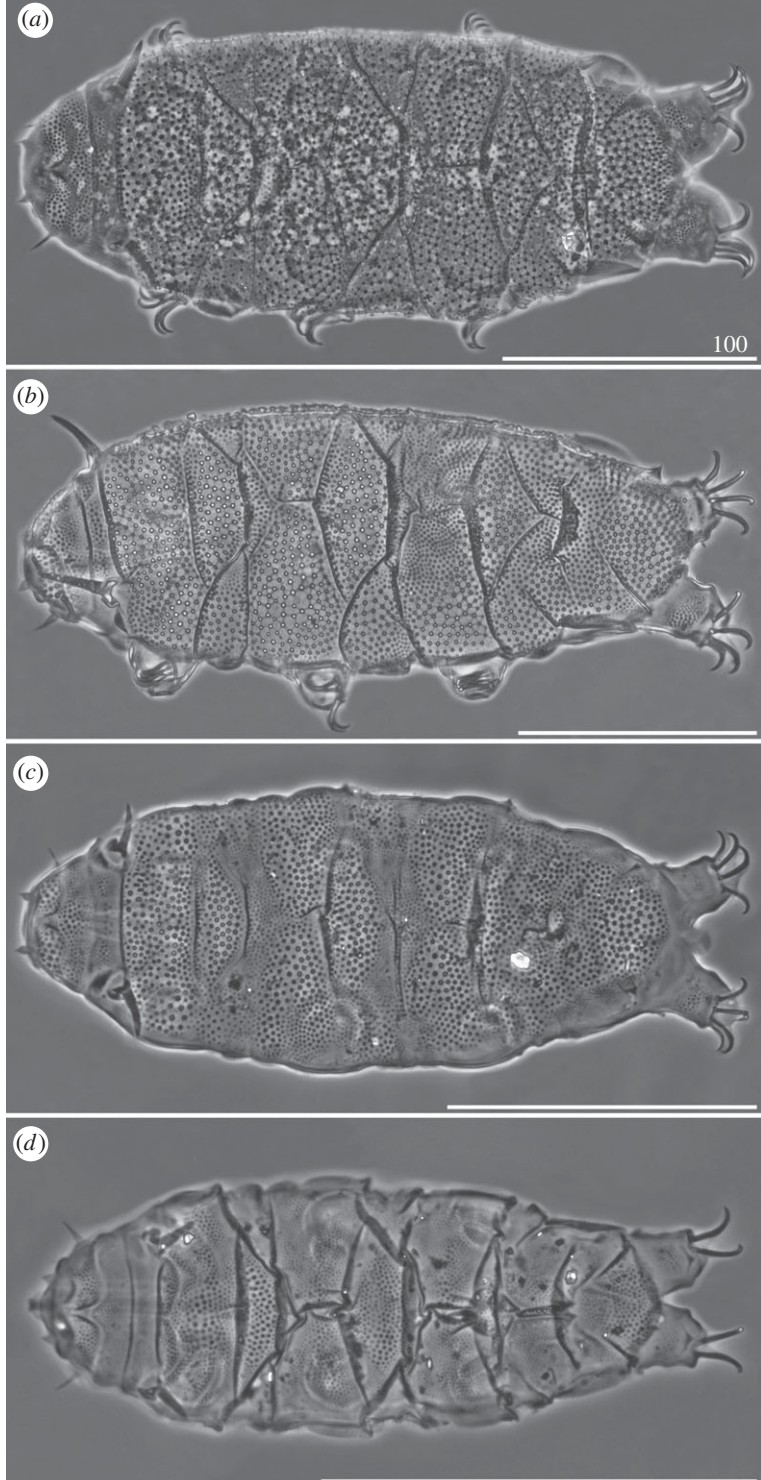

**Figure 9.** Habitus of *Cornechiniscus madagascariensis* (PCM): (*a*) female (dorsal view, holotype); (*b*) female (dorsolateral view, specimen from Ethiopia); (*c*) juvenile (dorsal view, specimen from Ethiopia); (*d*) larva (dorsal view, specimen from Ethiopia). All scale bars = 100 μm.

the *Echiniscus*-line, at a definitely more basal position with respect to the analyses by Kristensen [22], and *Hypechiniscus* as related to the *Echiniscus*-like genera.

The first molecular phylogeny of the Echiniscidae by Jørgensen *et al.* [19] demonstrated the diphyletic nature of *Pseudechiniscus*, gave support for the clade ((*Hypechiniscus* (*Echiniscus* + *Testechiniscus*)), and removed *Mopsechiniscus* from the *Pseudechiniscus*-line. Subsequently, a new genus *Diploechiniscus* was erected within this clade from *Echiniscus* [11], which, together with the improved sampling within the

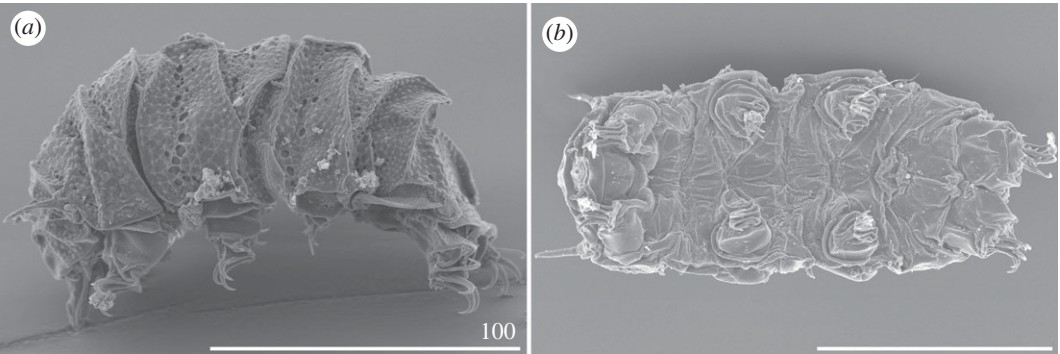

**Figure 10.** Habitus of *Cornechiniscus madagascariensis* (females from Ethiopia, SEM): (*a*) lateral view; (*b*) ventral view. All scale bars = 100 μm.

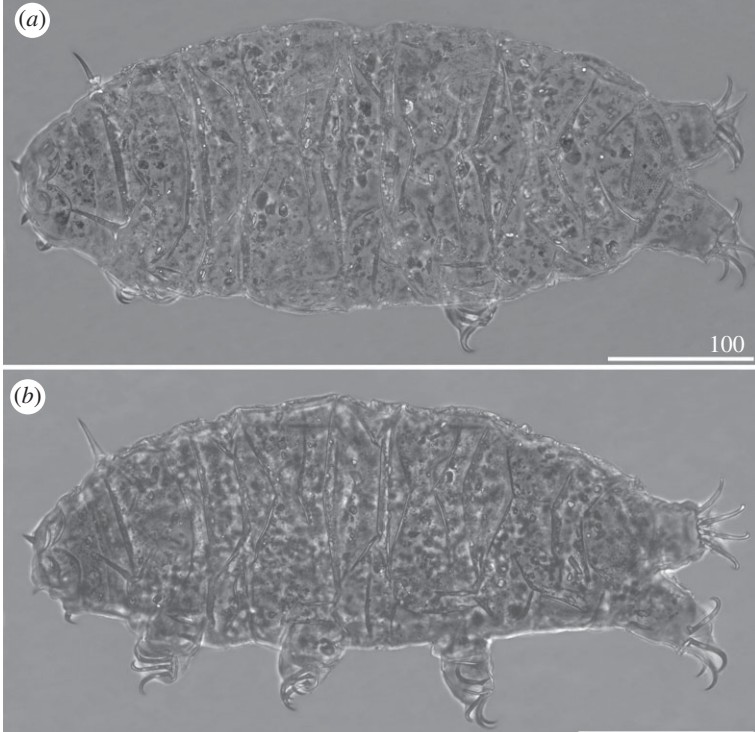

**Figure 11.** Habitus of *Cornechiniscus schrammi* (PCM): (*a*) female (dorsal view, paratype); (*b*) male (dorsolateral view, paratype). All scale bars = 100 μm.

speciose *Echiniscus* [16], underlined the importance of cuticular sculpture in the taxonomy of the group. The continuous efforts to uncover the biodiversity within Echiniscidae resulted in the formal splitting of *Pseudechiniscus* into two remotely related genera: *Pseudechiniscus* and *Acanthechiniscus* [5]. Also, the erection of *Stellariscus* and improved morphological sampling pinpointed the polyphyletic nature of *Testechiniscus*, and confirmed that *Bryodelphax* has more in common with the *Pseudechiniscus*-like genera than with the 'Echiniscus-line' [12]. Consequently, Gąsiorek [65] identified the *Echiniscus*-line and the *Bryodelphax*-line to stress the unique position of *Bryodelphax* on the phylogenetic tree of the Echiniscidae. Recently, a critical evaluation of dorsal sculpturing, ventral armature and development of trunk appendages as taxonomic traits, has led to the elevation of five new genera within the *Echiniscus*-line: *Barbaria*, *Claxtonia*, *Kristenseniscus*, *Nebularmis* and *Viridiscus* [20]. Another recent contribution [24] introduced new taxonomic ranks without an extensive discussion on morphology and with varying phylogenetic support. The phylogeny presented in Guil *et al.* [24] embraced both clades already widely recognized as monophyletic (the *Echiniscus*-line and the *Bryodelphax*-line *sensu* Gąsiorek [65]) and taxa with unresolved positions (the entire *Pseudechiniscus*-line). Moreover, the phylogenetic analysis presented in Guil *et al.* [24] only used published echiniscid sequences DNA

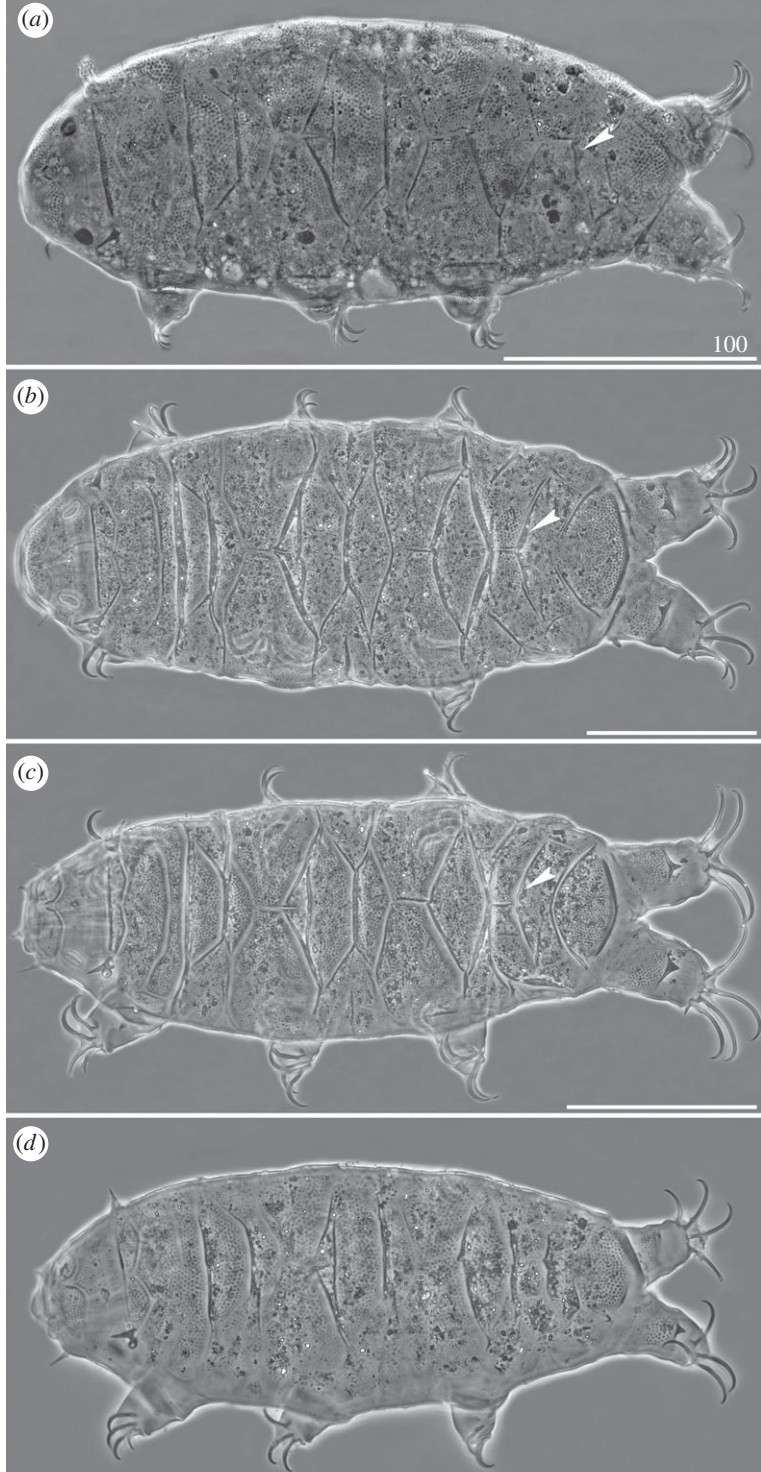

**Figure 12.** Habitus of *Cornechiniscus subcornutus* (PCM): (*a*) female (dorsolateral view, holotype); (*b*) female (dorsal view, specimen from Kyrgyzstan); (*c*) male (dorsal view, specimen from Kyrgyzstan); (*d*) juvenile (dorsolateral view, specimen from Kyrgyzstan). Arrowheads indicate weakly developed teeth at the posterior margin of the pseudosegmental plate IV′. All scale bars = 100 μm.

from earlier studies, thus providing no advancement in solving the phyletic relationships in this heterotardigrade family. Nevertheless, Guil *et al*. [24] proposed a subfamilial and tribal classification based on traditional, morphological groupings, which are not confirmed to be monophyletic by means of molecular phylogenetics. Furthermore, Guil *et al*. [24] did not take into consideration the morphological analyses by Jørgensen [23] and Gąsiorek *et al*. [12] that contradict the postulated

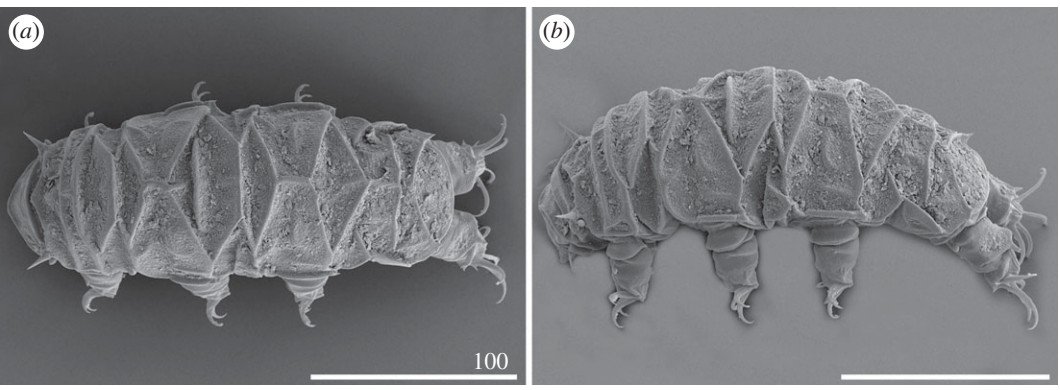

**Figure 13.** Habitus of *Cornechiniscus subcornutus* (females from Kyrgyzstan, SEM): (*a*) dorsal view; (*b*) lateral view. All scale bars = 100 μm.

**Figure 14.** Cephalic sensory organs of *Cornechiniscus* (all but (*a*) in SEM, (*a*) PCM): (*a,b*) *C. madagascariensis*; (*c,d*) *C. imperfectus* sp. nov. (Roman numerals signify segments of the mouth cone); (*e*) *C. lobatus*; (*f*) *C. madagascariensis*. Stars indicate weakly developed secondary clavae (cephalic papillae). Note bifurcated and trifurcated endings of peribuccal cirri (incised arrowheads). All scale bars in μm.

subfamilies and tribes. Therefore, in this paper we critically review the traits used by Guil *et al.* [24] to erect the subfamilies and tribes, and demonstrate them to be unreliable in the light of the new data for the genus *Cornechiniscus* presented in this study.

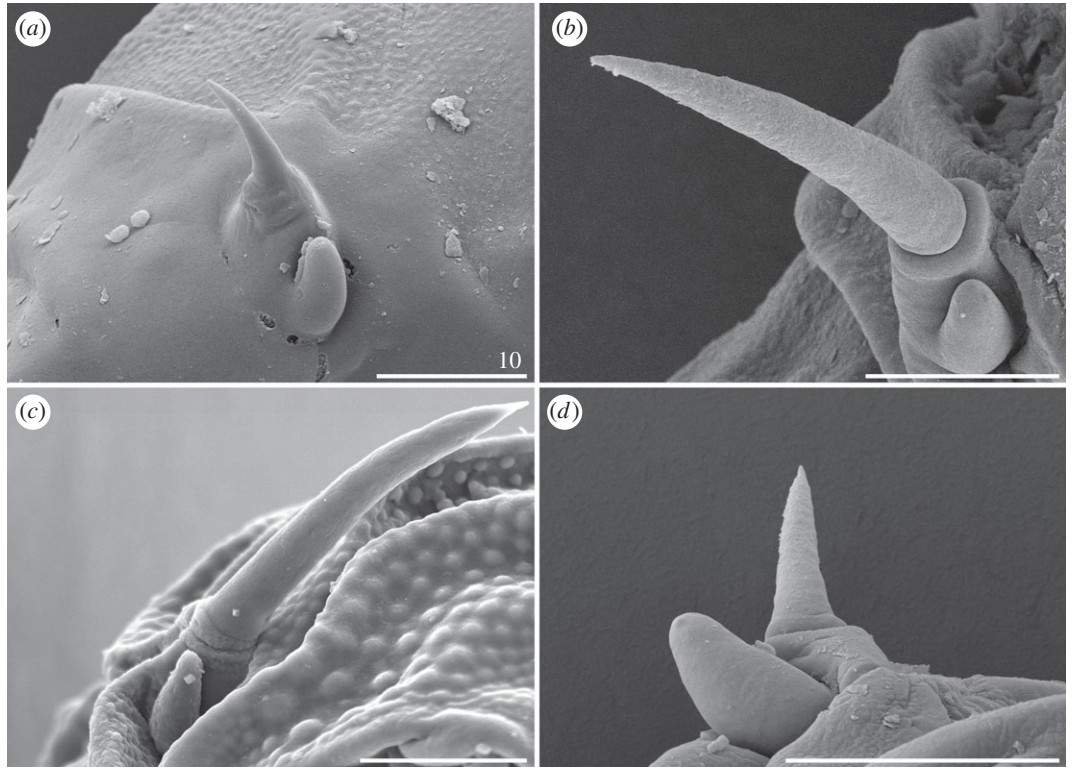

**Figure 15.** Modified cirrus *A* of *Cornechiniscus* (SEM): (*a*) *C. imperfectus* sp. nov.; (*b*) *C. lobatus*; (*c*) *C. madagascariensis*; (*d*) *C. subcornutus*. All scale bars = 10 μm.

The subfamily 'Echiniscinae' *sensu* Guil *et al*. [24] contains the tribe 'Bryodelphaxini' with *Bryochoerus* and *Bryodelphax* (the distinction between which is debatable and unsolved, see [22,66]), and 'Echiniscini' with *Diploechiniscus*, *Echiniscus*-like genera, *Hypechiniscus* and *Testechiniscus*. The genus *Bryodelphax* and its phylogenetic position remain of special significance for understanding the phyletic affinities between the echiniscid genera [22,23,65]. In the phylogenetic analysis presented herein, *Bryodelphax* constitutes the sister taxon to all remaining echiniscid genera (figure 1), which invokes the analyses by Jørgensen [23], clearly indicating the 'basal' character of this genus within the clade of echiniscid genera with paired segmental plates. This explicitly demonstrates that the alleged synapomorphy of putative 'Echiniscinae', i.e. the lack of pseudosegmental plates, is a convergent homoplasy, meaning that the subfamily Echiniscinae is diphyletic and artificial. Moreover, the trait used to delimit 'Bryodelphaxini' from 'Echiniscini', namely the absence of cirrophores in the peribuccal cirri of *Bryodelphax*, is incorrect as the peribuccal cirrophores are present in *Bryodelphax* (see Maucci [67, fig. 91] for *B. parvulus*, Kristensen *et al*. [68, fig. 21] for *B. aaseae*, Gąsiorek *et al*. [69, fig. 12] for *B. maculatus*). The miniaturization of *Bryodelphax* [65] entailed a simplification of some aspects of its anatomy. For example, the peribuccal cirrophores exhibit various stages of reduction (i.e. merging with flagellum), but this is also the case for other genera, such as *Cornechiniscus* (figure 14), meaning that there is no morphological support for the distinction between these two tribes, consisting of two distant lineages.

Another genus with an uncertain phylogenetic position is *Mopsechiniscus*. In the analysis in which DNA sequences for this genus were published for the first time [19], the phylogenetic position of *Mopsechiniscus* was unsolved. However, in Guil *et al*. [24], *Mopsechiniscus* was inferred as sister to all remaining echiniscids. Nevertheless, Guil *et al*. [24] placed this taxon in 'Pseudechiniscini' together with the distantly related *Pseudechiniscus*, making the tribe polyphyletic at the time of its erection. In the present study, *Mopsechiniscus* is not the most basal echiniscid genus, but it remains a remote kin of *Pseudechiniscus*. Despite the obvious polyphyly, Guil *et al*. [24] grouped *Mopsechiniscus* with *Pseudechiniscus* within the 'Pseudechiniscini' based on the presence of the pseudosegmental plate IV'. However, as demonstrated above, the same arrangement of dorsal plates may characterize genera from different phyletic lineages, thus this trait cannot be used at a suprageneric level. Additionally, many details of the anatomy of the bucco-pharyngeal apparatus of *Mopsechiniscus* are much closer to *Cornechiniscus* and *Proechiniscus*, as noted by Kristensen [22] and confirmed by Guidetti *et al*. [70], than to *Pseudechiniscus s.s.* which exhibits

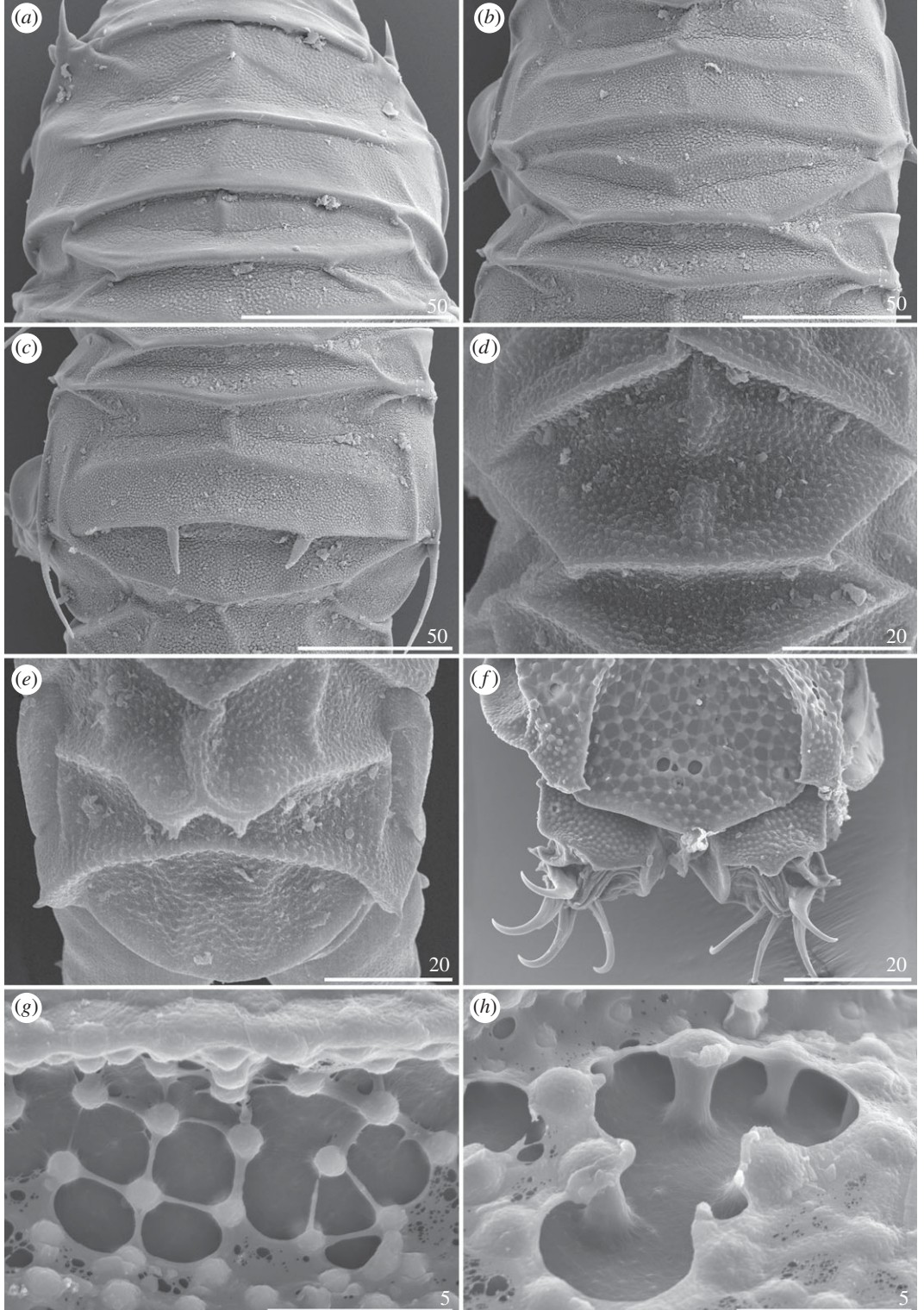

**Figure 16.** Dorsal plate sculpturing of *Cornechiniscus* (SEM): (a–c) *C. imperfectus* sp. nov.; (d,e) *C. lobatus*; (f–h) *C. madagascariensis*. All scale bars in µm.

a simple, plesiomorphic apparatus. Also, the cushion-like organs on legs indicate a closer relationship between *Mopsechiniscus* and *Acanthechiniscus* + *Cornechiniscus* ( = members of the putative 'Cornechiniscini'; [14,70]) than between *Mopsechiniscus* and *Pseudechiniscus*. In short, the morphological evidence for the kinship of *Mopsechiniscus* and *Pseudechiniscus* is poor. In the phylogenetic analysis presented herein, *Mopsechiniscus* is in polytomy with *Parechiniscus*, the *Echiniscus* lineage and the 'Cornechiniscini' (figure 1), which is in agreement with the recent analyses by Cesari *et al*. [17] under the

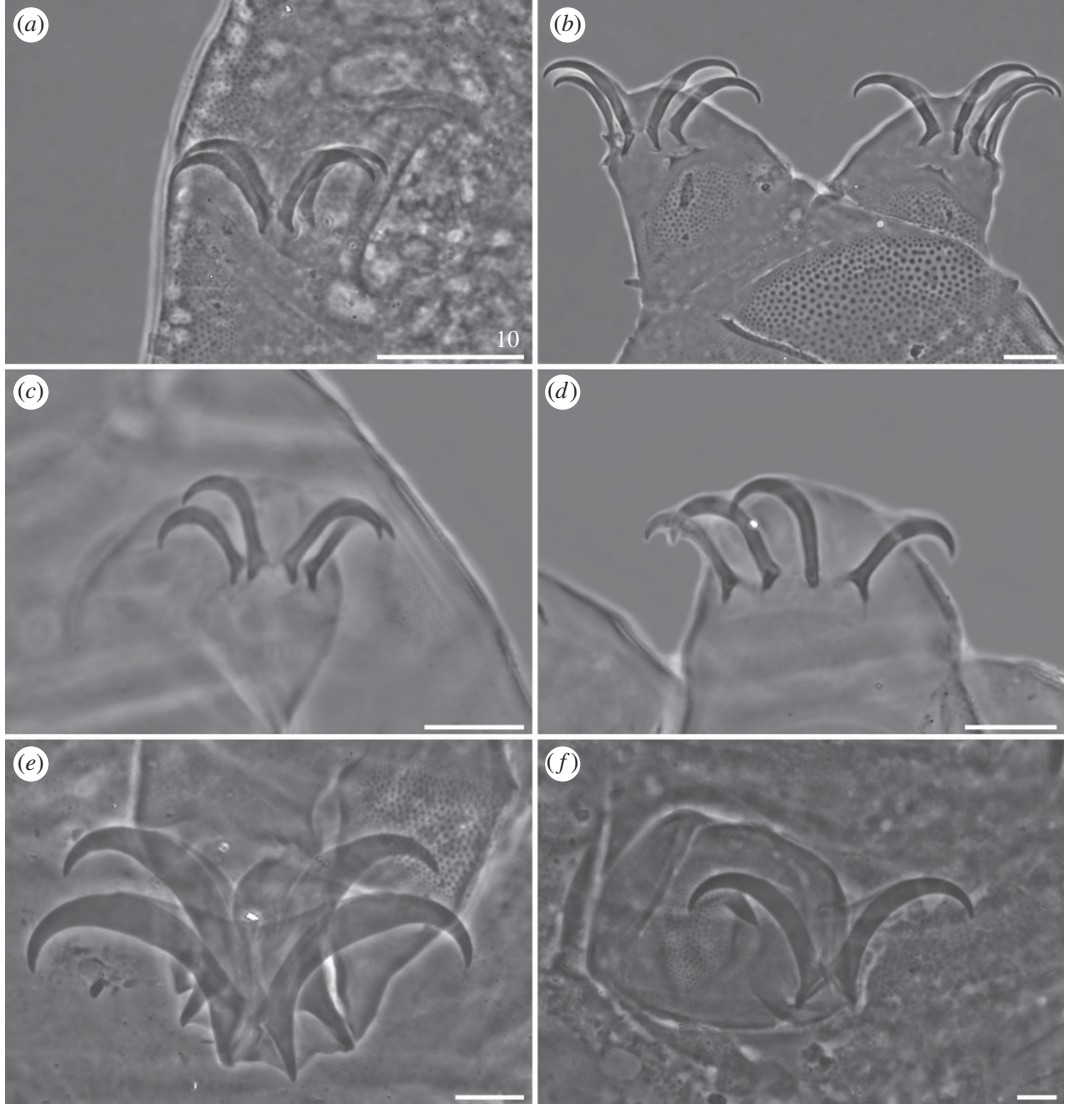

**Figure 17.** Claw morphology of *Cornechiniscus* (PCM): (*a*) *C. brachycornutus* (claws III); (*b*) *C. brachycornutus* (claws IV); (*c*) *C. cornutus* (claws II); (*d*) *C. cornutus* (claws IV); (*e*) *C. holmeni* (claws III); (*f*) *C. holmeni* (larval claws I). All scale bars = 10 μm.

evolution model from the GTR family, i.e. the model returning phylogenies with the best-resolved nodes and congruent with morphological clues (e.g. [20] or [71]). The position of this genus will probably not be resolved until genetic data are acquired for more *Mopsechiniscus* spp., *Antechiniscus* spp., and, crucially, *Novechiniscus*. The clade ((*Proechiniscus* (*Acanthechiniscus* + *Cornechiniscus*)) ( = 'Cornechiniscini') inferred in the present contribution (figure 1) is congruent with the recent analyses [5,12,17,24], and, consequently, is the second monophyletic clade after the earlier recognized *Echiniscus* lineage [20,22]. However, the trait used by Guil *et al.* [24] to separate 'Cornechiniscini' from other 'Pseudechiniscinae'—the presence of pseudosegmental plates I' and III'—is not a state present in *Acanthechiniscus* [5] and, actually, varies within *Cornechiniscus* itself, as there are species in which all pseudosegmental plates I–IV' are developed (*C. holmeni*, *C. imperfectus* sp. nov.), but also species with only pseudosegmental plates I' and IV' exist (*C. lobatus*, *C. madagascariensis*, *C. subcornutus*). This variability renders the presence of pseudosegmental plates I'–III' as not useful in inferring phylogeny at a supra-generic level. What is more, the presumptive 'Antechiniscini', distinguished by the presence of pseudosegmental plates II'–IV', exhibit no distinct synapomorphy. They are a mix of two unrelated genera: (i) *Antechiniscus*, sharing the majority of available morphological criteria with *Pseudechiniscus* ([13], hence the inferred clade *Antechiniscus* + *Pseudechiniscus* in the morphology-based phylogeny from [12]), and (ii) *Multipseudechiniscus*, having all pseudosegmental plates I–IV', as some *Cornechiniscus* spp. Moreover, the bucco-pharyngeal apparatus of *Multipseudechiniscus* closely resembles the apparatus of *Cornechiniscus* [72]. In result, there is no pivotal difference between 'Cornechiniscini' and 'Antechiniscini' *sensu* Guil *et al.* [24].

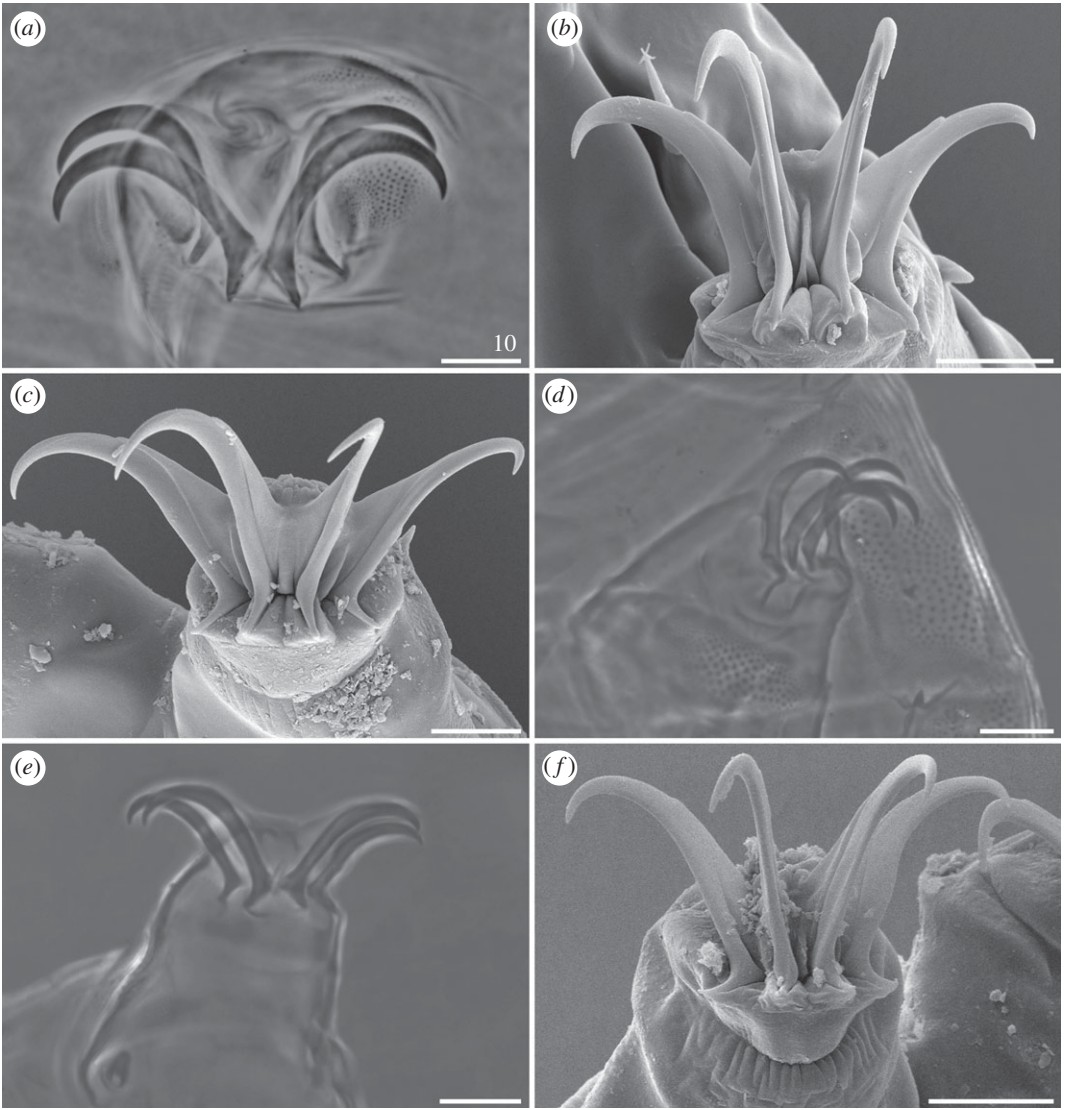

**Figure 18.** Claw morphology of *Cornechiniscus*: (*a*) *C. imperfectus* sp. nov. (claws I, PCM); (*b*) *C. imperfectus* sp. nov. (claws I, SEM); (*c*) *C. imperfectus* sp. nov. (claws IV, SEM); (*d*) *C. lobatus* (claws II, PCM); (*e*) *C. lobatus* (claws IV, PCM); (*f*) *C. lobatus* (claws IV, SEM). All scale bars = 10 µm.

All morphological phylogenies published so far inferred *Parechiniscus* and *Novechiniscus* as the 'basal' genera of the Echiniscidae [12,22,23]. Although genetic data are yet to be obtained for *Novechiniscus*, all morphological phylogenies returned a step-like arrangement of echiniscid genera. This seems logical from an evolutionary point of view since the sclerotization of dorsum, and subsequent complication of its anatomy, in the Echiniscidae was most likely gradual (i.e. in the sequence *Parechiniscus*-like echiniscids → *Novechiniscus*-like echiniscids → 'advanced' genera with paired segmental plates). Guil *et al*. [24] united both *Parechiniscus* and *Novechiniscus* into 'Parechiniscinae', which, according to earlier morphological analyses, are most probably paraphyletic with respect to the *Echiniscus*-line and the *Pseudechiniscus*-line [12,22,23]. Furthermore, the feature used as a synapomorphy for this subfamily, i.e. the absence of the cervical (neck) plate, is dubious, as this element has only recently been used in echiniscid species descriptions and seems to be variable within genera (e.g. [73]). Consequently, the diphyly of 'Echiniscinae', the polyphyletic character of 'Pseudechiniscinae', and the most probable paraphyletic character of 'Parechiniscinae', together with ambiguous 'apomorphies' of the tribes proposed in Guil *et al*. [24], incline us to consider this hypothetical classification as unfounded (see table 12 for the summary of phylogenetic and morphological evidence against the subfamilial and tribal classification). In result, all echiniscid subfamilies and tribes postulated by Guil *et al*. [24] are designated as invalid. Considering the still limited molecular dataset for echiniscids, the notoriously unstable phylogenetic positions of some pivotal genera (e.g. *Bryodelphax*, *Mopsechiniscus*), and

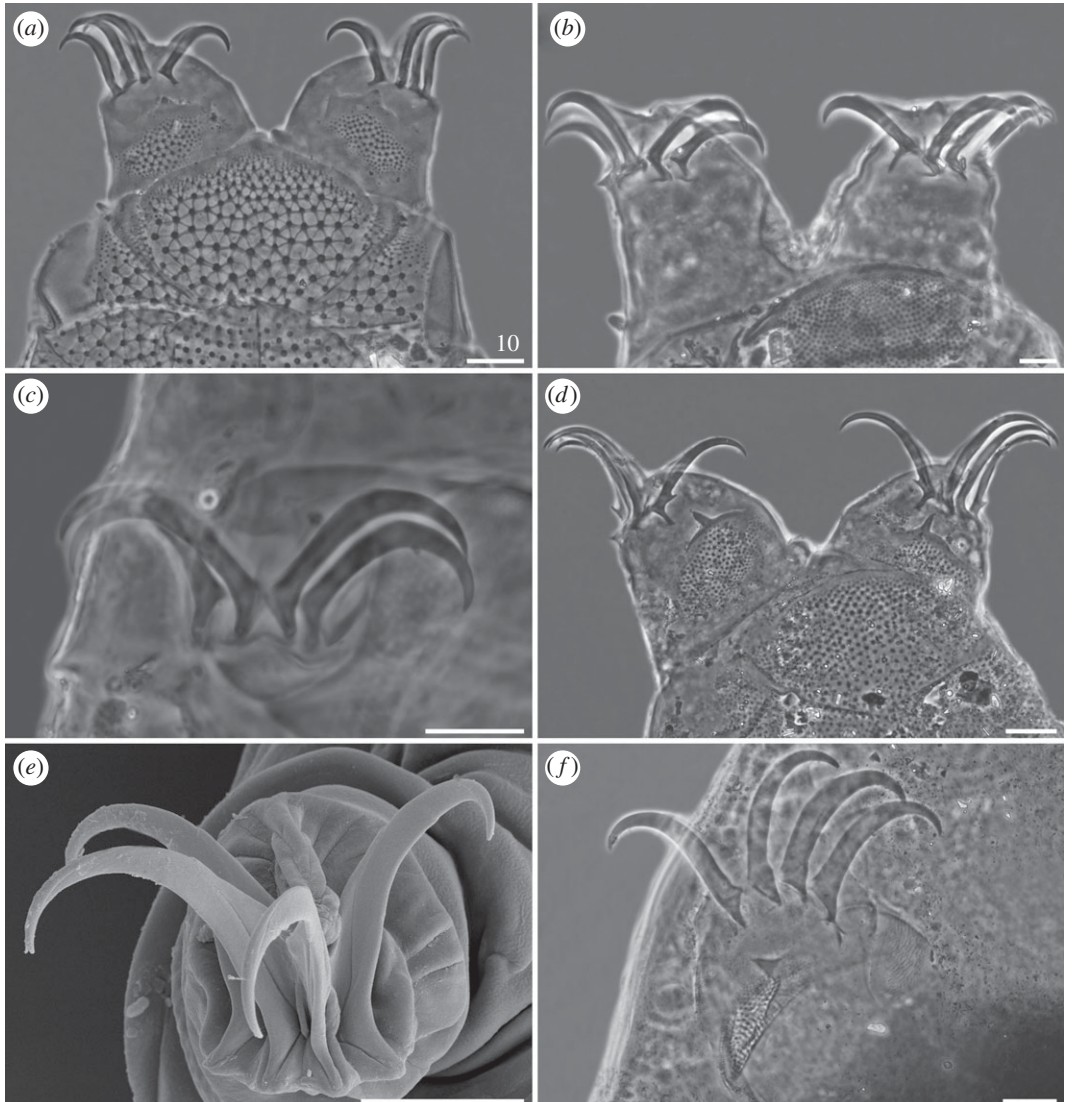

**Figure 19.** Claw morphology of *Cornechiniscus*: (*a*) *C. madagascariensis* (claws IV, PCM); (*b*) *C. schrammi* (claws IV, PCM); (*c*) *C. subcornutus* (claws II, PCM); (*d*) *C. subcornutus* (claws IV, PCM); (*e*) *C. subcornutus* (claws III, SEM); (*f*) *C. tibetanus* (claws I, PCM). All scale bars = 10 µm.

incongruences resulting from choosing different evolution models, none of the currently available echiniscid phylogenies are conclusive and they do not allow for the establishment of subfamilies and/ or tribes.

## 4.2. Synapomorphies and other characteristics of the *Cornechiniscus* clade

The clade comprising *Acanthechiniscus*, *Cornechiniscus* and *Proechiniscus* has been demonstrated in Vecchi *et al.* [5], and later supplied with *Multipseudechiniscus* by Gąsiorek *et al.* [12]. The morphological uniformity of this group is well supported as there are several traits common to the four genera: large, oval, black crystalline eyes [5,22,72], the reduction and miniaturization of primary clavae [22], and the lack of ventral armature [12]. Moreover, there are traits that are shared by either three or two of these genera. Specifically, the bucco-pharyngeal apparatus with a long buccal tube, strengthened by crests slightly before the posteriormost, thinned flexible portion is characteristic for *Cornechiniscus*, *Proechiniscus* and *Multipseudechiniscus*. At the same time, long, filamentous lateral cirri in at least one trunk position E unify *Acanthechiniscus*, *Proechiniscus* and *Multipseudechiniscus* (figure 22). Furthermore, peribuccal *cirri* are bifurcated or tufted at their tips in *Acanthechiniscus* and *Cornechiniscus* ([74,75]; figure 14). Finally, *Acanthechiniscus*, *Cornechiniscus* and *Multipseudechiniscus* contain species with a body size significantly larger (greater than 300 µm, usually much longer) than is typical for the majority of the Echiniscidae (less

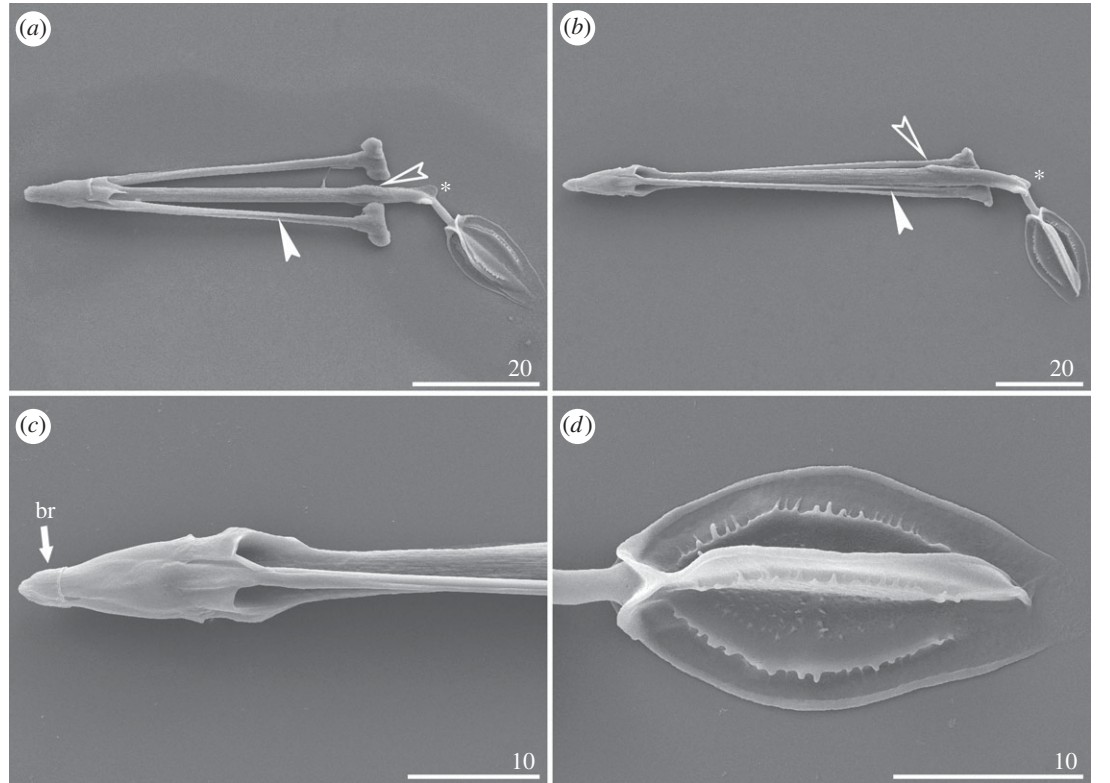

**Figure 20.** Bucco-pharyngeal apparatus of *Cornechiniscus* (SEM): (*a*) *in toto* (*C. lobatus*); (*b*) *in toto* (*C. imperfectus* sp. nov.); (*c*) close-up on the buccal cone and stylet sheaths (*C. imperfectus* sp. nov.); (*d*) pharynx (*C. imperfectus* sp. nov.). Explanation of symbols: br, buccal ring; incised arrowheads—longitudinal stylet grooves, empty incised arrowheads—dorsal crest, stars—lateral crest. All scale bars in µm.

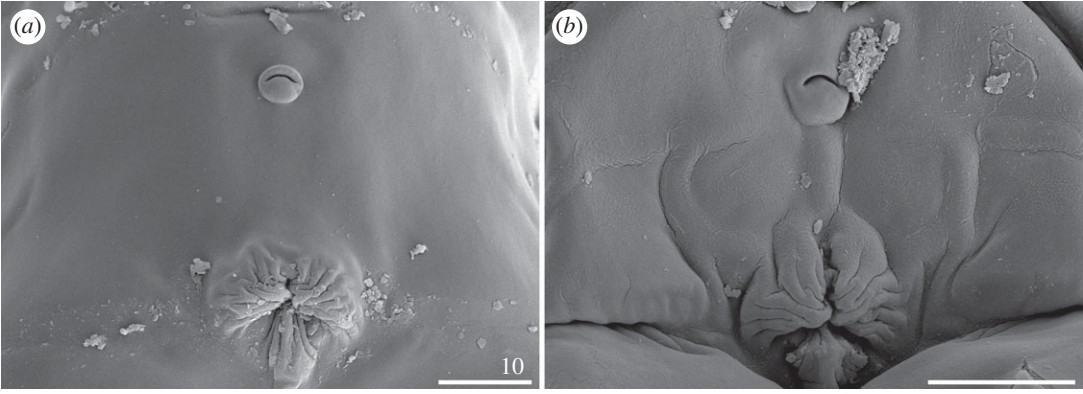

**Figure 21.** Male gonopore of *Cornechiniscus* (SEM): (*a*) *C. imperfectus* sp. nov.; (*b*) *C. subcornutus*. All scale bars = 10 µm.

than 250 µm). This clade is also interesting for an another reason, specifically because of different patterns of modes of reproduction among the genera: *Proechiniscus* and *Multipseudechiniscus*, currently monotypic, are dioecious, but the polytypic *Acanthechiniscus* is probably parthenogenetic (males were never found) whereas *Cornechiniscus* most likely represents a mixture of reproductive modes, as males were reported only in 3 of the 10 known species (thus, the remaining seven species are probably parthenogenetic).

The species *Acanthechiniscus goedeni* [34] was listed as requiring further examination to confirm whether it represents *Acanthechiniscus* or *Multipseudechiniscus* [5]. The PCM analysis of a paratype of this taxon indicates its affinity to *Acanthechiniscus* (figure 22*b*), as it has a paired pseudosegmental plate IV′ and thick cirri in all lateral and in a single dorsal positions (body appendage formula A-B-C-D-D$^d$-E). Moreover, dorsal plate sculpturing in *A. goedeni* (two kinds of pillars: large, widely spaced, and not forming any shapes, and small, densely arranged, which results in ridge-like extensions on the epicuticle; figure 22*b*) is similar to that in *A. islandicus* [76], in which densely packed endocuticular pillars form an autapomorphic pattern resembling a reticulum.

**Table 12.** The summary of evidence against the subfamilial and tribal classification of Echiniscidae proposed by Guil *et al.* [24]. Two tribes, Cornechiniscini and Parechiniscini, are the only taxa that are monophyletic and exhibit autapomorphies; however, they cannot be preserved when all other taxa are invalid.

| taxon | genera (alphabetically) | main problem |
|---|---|---|
| Echiniscinae | *Barbaria, Bryochoerus, Bryodelphax, Claxtonia, Diploechiniscus, Echiniscus, Hypechiniscus, Kristenseniscus, Nebularmis, Stellariscus, Testechiniscus, Viridiscus* | diphyly |
| Echiniscini | *Barbaria, Claxtonia, Diploechiniscus, Echiniscus, Hypechiniscus, Kristenseniscus, Nebularmis, Stellariscus, Testechiniscus, Viridiscus* | lacking an autapomorphy |
| Bryodelphaxini | *Bryochoerus, Bryodelphax* | lacking an autapomorphy |
| Pseudechiniscinae | *Acanthechiniscus, Antechiniscus, Cornechiniscus, Mopsechiniscus, Multipseudechiniscus, Proechiniscus, Pseudechiniscus* | polyphyly<br>pseudosegmental plates are plesiomorphic and their number vary within a genus |
| Pseudechiniscini | *Mopsechiniscus, Pseudechiniscus* | diphyly<br>lacking an autapomorphy |
| Cornechiniscini | *Acanthechiniscus, Cornechiniscus, Proechiniscus* | — |
| Antechiniscini | *Antechiniscus, Multipseudechiniscus* | diphyly<br>pseudosegmental plates II'–IV' are present also in *Cornechiniscus* and *Proechiniscus* |
| Parechiniscinae | *Novechiniscus, Parechiniscus* | paraphyly |
| Parechiniscini | *Parechiniscus* | — |
| Novechiniscini | *Novechiniscus* | lacking an autapomorphy |

## 4.3. Morphology of *Cornechiniscus*

The bulk of knowledge about the morphological variation within *Cornechiniscus* comes from the works of Maucci [27,33], who described as many as 5 of the 10 currently known species in the genus and noted the divergence between the *Pseudechiniscus cornutus* group (species grouped under the current diagnosis of *Cornechiniscus*) and the *Pseudechiniscus victor* group (the recently erected *Acanthechiniscus*). With the broader sampling and modern taxonomic tools, it is possible to describe some novel findings that are important for the classification of the genus. Abe & Takeda [77] noted for the first time that *C. madagascariensis* exhibits ventral cuticular grooves. Interestingly, we observed that such a regular wrinkling exists in all species devoid of trunk cirri whereas the two known appendaged *Cornechiniscus* species, *C. holmeni* and *C. imperfectus* sp. nov., have smooth abdomens. Also, the present study shows that the reduction and miniaturization of *cirri A* happened at least twice in the course of evolution of the genus, as species with short and poorly developed cephalic appendages, *C. imperfectus* sp. nov. and *C. subcornutus* (figure 15*a,d*), are not directly related (figure 2).

Our phylogenetic analyses based on the currently available dataset for the genus *Cornechiniscus* showed that *C. lobatus* and *C. madagascariensis* are sister species (figure 2), which is in agreement with earlier opinions about the close affinity of these two species [31,77]. The two species are most easily differentiated by the presence of prominent *striae* in *C. madagascariensis* which—according to the original description and our observations of syntypes (in LCM) and new individuals (under LCM and SEM)—are absent in *C. lobatus* (figure 16*d–e*). However, it should be mentioned that Abe & Takeda [77] reported *striae* in the same syntypes of *C. lobatus* as we examined for this study. This discrepancy stems from a different interpretation of what constitutes *striae*. Currently, *striae* are defined as evident, thin, linear connections between spaced cuticular pillars [12,18], such as these observed in

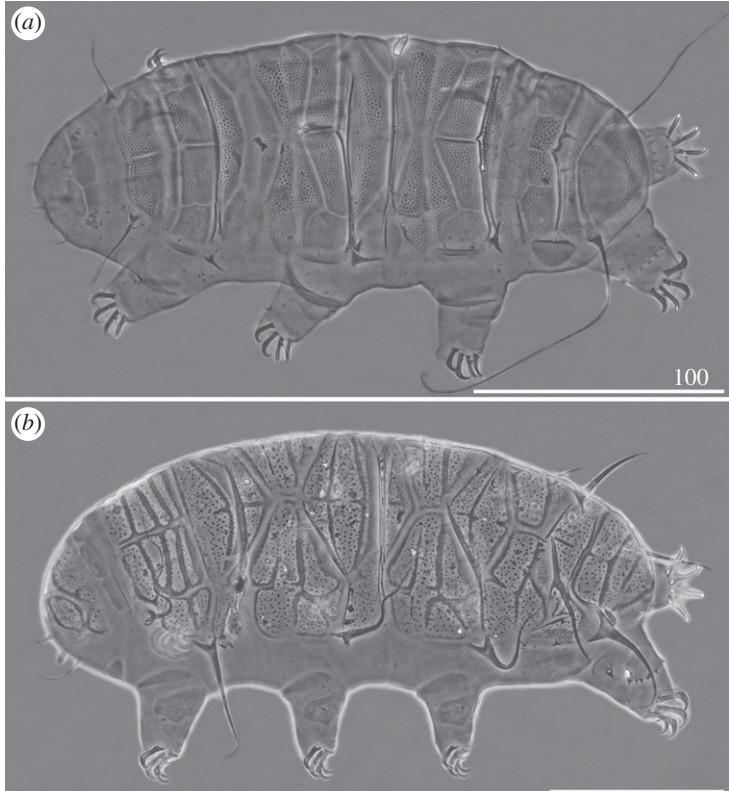

**Figure 22.** The closest relatives of *Cornechiniscus* (PCM): (*a*) *Multipseudechiniscus raneyi*; (*b*) *Acanthechiniscus goedeni*. All scale bars = 100 μm.

*C. madagascariensis* (figure 16*f–h*). In *C. lobatus*, however, small polygonal pillars are closely arranged and in direct contact by their corners (figure 16*d,e*), which are not *striae*, but were interpreted as such in [77]. Therefore, large, widely spaced endocuticular pillars with prominent *striae* in *C. madagascariensis* remain the most important discriminating character to separate from *C. lobatus* (see also the key below).

The three types of relationship of claw lengths described in the present contribution (i.e. isonych, slightly heteronych (anisonych) and strongly heteronych) seem to hold no phylogenetic signal, as the sister species to an isonych *C. cornutus* (figure 17*c,d*) is a strongly heteronych *C. imperfectus* sp. nov. (figures 2 and 18*a–c*). Thus, this criterion useful for species delineation cannot be treated as a potential autapomorphy for subclades within *Cornechiniscus*.

The bucco-pharyngeal apparatus of *C. lobatus* was analysed in SEM and described in detail by Guidetti *et al.* [78] although the specimen was in a poor condition, i.e. overdigested and not fully cleansed (the apparatus is easily deformable during extraction due to its delicate anatomy; [22]). This may be the cause of a misinterpretation that the longitudinal dorsal crest is followed by a second longitudinal crest [78]. Our observations, made with a good quality specimen, unequivocally show that the longitudinal dorsal crest is followed by two lobe-like, lateral crests (figure 20*a,b*), which resembles the state in *Mopsechiniscus* that exhibits two massive lateral thickenings [79].

## 4.4. Biogeography of *Cornechiniscus*

Kristensen [22] stated that the genus is a typical Palaearctic element but numerous subsequent records from other regions of the world reported in the literature [27,31,80] falsified this statement. Central Asian mountain ranges (Tien-Shan, the Himalayas, Karakorum), valleys and mountainous plateaus harbour the highest diversity of *Cornechiniscus* species. In the present work, fours species are recorded from mountains in Kyrgyzstan (*C. cornutus*, *C. imperfectus* sp. nov., *C. lobatus*, *C. subcornutus*). This is the second case of this many *Cornechiniscus* spp. exhibiting sympatric ranges. Kristensen [22] reported *C. cornutus* and *C. holmeni* from Kashmir, but there are two other species inhabiting this region: *C. lobatus* and *C. madagascariensis* (confirmed after a re-analysis of the material deposited in Copenhagen for the purpose of this study). Adding single reports of *C. schrammi* from Afghanistan [32] and of *C. tibetanus* from the Himalayas [33], it

appears that 80% of currently recognized species occur in dry habitats at high elevations in Central Asia. Biserov [81] provided records of *C. cornutus*, *C. schrammi* and *C. tibetanus* from mountains of Turkmenistan. It has been shown that biodiversity hotspots constitute areas of diversification and speciation for certain group of animals (e.g. [82,83]), thus the presence of the vast majority of *Cornechiniscus* spp. in the Central Palaearctic implies that this could be the region where the genus evolved. Moving towards the Iberian Peninsula, only four species dwell in the Western Palaearctic (*C. cornutus*, *C. lobatus*, *C. subcornutus* and *C. holmeni*, but the latter only on the highest mountain peaks).

Four species, *C. holmeni*, *C. lobatus*, *C. madagascariensis* and *C. subcornutus*, have particularly interesting geographical ranges from the perspective of testing how the genus disperses and colonizes new areas. Specifically, *C. holmeni* was originally described from Greenland [29], and was commonly considered a rare species [84], but later reports from the Caucasus [85], the Canadian Arctic [86,87], Italian Alps [67,84], Kashmir [22], Mongolia [88] and the Chinese part of Tien-Shan [89], elucidated its stenothermic preferences, restricting the occurrence to single insular habitats dispersed across the Holarctic. A similar biogeographic pattern was suggested for *Eohypsibius* [90], *Bertolanius* [91] and recently for *Cryoconicus* ([92], later amended by the data from Antarctica, see [93]). Furthermore, *C. lobatus* has the broadest geographical range of all known congeners, being present in entire temperate Eurasia, Mediterranean Africa, North America and northern parts of South America [80,94–96]. This suggests a great dispersal potential of this taxon, and probably the colonization of the northern Neotropic from the Nearctic [97]. However, the relatively wide geographical range of *C. lobatus* may also be explained alternatively by a complex of cryptic or pseudocryptic species. Thus, the two hypotheses need to be tested with the use of DNA barcodes (e.g. [46]). By contrast, *C. madagascariensis* was recorded so far only from its locus typicus, Himachal Pradesh (India) and Reunion [31,77,98]. However, the examination of slides deposited in Copenhagen revealed its presence also in southwestern Kenya (Gembe Hills, expedition 1987–1988), northern Tanzania, Ethiopia (expedition 2016) and Kashmir (see also above), which means that the distribution of *C. madagascariensis* may cover the entire Indian Ocean basin. Finally, *C. subcornutus*, with only two reports, the original record from the Iberian Peninsula and the Kyrgyzstan locality described herein, seems to have a disjunctive range. The species was described based on the finding of five females, with the posterior margin of the pseudosegmental plate IV′ terminating with two teeth [25]. The comparison of the type material with the freshly obtained population from Kyrgyzstan pinpointed that in both populations, the majority of specimens exhibited minute and barely distinguishable teeth (figure 12) rather than the large dentate projections described originally by Maucci & Ramazzotti [25]. Furthermore, the arrangement and level of sclerotization of dorsal plates, the type of sculpture, and extraordinarily heteronych claws are shared by both populations. Consequently, there is no morphological evidence allowing for the delimitation of European and Asian populations. The ultimate test of the identity of the Kyrgyz population requires topotypic DNA sequences from Spain. In fact, we analysed samples from the type locality of *C. subcornutus* in the vicinity of Huesca, but we did not find the species (although, for example, *Tenuibiotus ciprianoi* [99], another xerophilous taxon, was present in the samples). In result, the disjunctive distribution cannot be rejected until DNA sequences are collected for *C. subcornutus* from the locus typicus.

## 4.5. Key to *Cornechiniscus* spp.

The following taxonomic key refers to adults (both females and males), as sexually immature life stages (i.e. larvae and juveniles) have not yet been found for all *Cornechiniscus* species.

1. Filamentous lateral *cirri* C and D present   ................................................................................................... 2
-. Filamentous lateral *cirri* C and D absent   ................................................................................................. 3
2(1). Males absent, body appendage configuration $A$-$C$-$D$-$(D^d)$-$(ps)$-$E$, cirrus $A$/body length ratio 6–8% ............................................................................................................................................**C. holmeni**
-. Males present, body appendage configuration: $A$-$C$-$D$-$D^d$-$E$, cirrus $A$/body length ratio 3–4% ............
................................................................................................................................**C. imperfectus** sp. nov.
3(1). Claws isonych and miniaturized   ..........................................................................................**C. cornutus**
-. Claws heteronych and elongated    ...................................................................................................... 4
4(3). Legs IV with the dentate collar consisting of 3–7 teeth; teeth in the position $B^d$ present   .................
..............................................................................................................................................**C. ceratophorus**
-. Legs IV with the dentate collar consisting of 1–2 teeth or lacking dentate collar; teeth in the position $B^d$ absent    ................................................................................................................................................... 5
5(4). Dentate collar IV and tooth on legs IV absent ...............................................................**C. schrammi**

-. Dentate collar IV or tooth on legs IV present ............................................................. 6

6(5). *Cirri A* at least two times longer than cirri externi ........................................... 7

-. *Cirri A* of a similar length as cirri externi or shorter ........................................... 8

7(6). Endocuticular pillars very large on the central portions of dorsal plates (Ø: 2–2.5 μm) and connected by thick *striae* ...................................................................*C. madagascariensis*

-. Endocuticular pillars small to medium-sized on the central portions of dorsal plates (Ø: 1–1.5 μm) and without *striae* ........................................................................................*C. lobatus*

8(6). A single tooth IV present, males present .................................................*C. subcornutus*

-. Two teeth IV present, males absent ........................................................................ 9

9(8). Teeth IV long and narrow, pseudosegmental teeth massive and long ............................*C. tibetanus*

-. Teeth IV short and broad, pseudosegmental lobe with two spicule-like apices ......................................
................................................................................................................*C. brachycornutus*

Ethics. The expedition to Kyrgyzstan and fieldwork was carried out within the frame of the project no. DS/MND/WB/IZ/16/2018.

Data accessibility. The DNA sequences are deposited in GenBank, and morphometric spreadsheets are accessible on Tardigrada Register (www.tardigrada.net/register). The datasets supporting the conclusions of this article are included within the article, and concatenated sequence datasets are provided as electronic supplementary material.

Authors' contributions. P.G. and Ł.M. conceived the study, analysed data and wrote the manuscript. Both authors finally approved the manuscript for publication.

Competing interests. We declare we have no competing interests.

Funding. The study was supported by grants from the European Commission's Integrated Infrastructure Initiative programme SYNTHESYS (grant no. DK-TAF-6332 to P.G.) and from the Polish National Science Centre (grant *Preludium* no. 2019/33/N/NZ8/02777 to P.G., supervised by Ł.M.). Some analyses were performed with equipment purchased from the *Sonata Bis* programme of the Polish NSC (grant no. 2016/22/E/NZ8/00417 to Ł.M.).

Acknowledgements. Witold Morek and Bartłomiej Surmacz (Jagiellonian University) led the expedition to Kyrgyzstan in September 2018, during which the bulk of the material needed to accomplish this work was collected, thus they are most sincerely thanked for the effort and help. Reinhardt M. Kristensen, Asger Ken Pedersen (University of Copenhagen, Denmark) and Roberto Guidetti (University of Modena and Reggio Emilia, Italy) are gratefully acknowledged for making the collections of Natural History Museums of Denmark and Verona available, and for sharing precious moss samples or information for the purposes of this revision. We also thank Daniel Stec (Jagiellonian University) who prepared the material from Ethiopia during his stay in Copenhagen under the framework of the European Commission's programme SYNTHESYS (grant no. DK-TAF-5609 to D.S.). We are also grateful to Reinhardt M. Kristensen and an anonymous reviewer for their valuable comments, and to Sandra McInnes (British Antarctic Survey) who kindly proof-read the manuscript.

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
