## [Reviewer comments · Royal Society Open Science]

Review History

RSOS-200581.R0 (Original submission)

Review form: Reviewer 1

Is the manuscript scientifically sound in its present form?

Yes

Are the interpretations and conclusions justified by the results?

Yes

Is the language acceptable?

Yes

Do you have any ethical concerns with this paper?

No

Have you any concerns about statistical analyses in this paper?

No

Recommendation?

Accept with minor revision (please list in comments)

Comments to the Author(s)

General comments

The work is well written, the critical part on previous phylogeny data is well supported and should be accepted, the descriptions are accurate, there is a lot of morphometry and the images are particularly beautiful and demonstrative.

My suggestion is for minor changes:

1. Topology of Fig. 1B is not sufficiently supported and could create misunderstanding. My suggestion is to delete the values under 70%.
2. In the description of the amendments a reference to the original description is lacking. As a consequence, it is not clear what was contained in the original description, what added and what changed. In several cases the original description is well done, not in other cases. Considering that the authors have examined the type material of several species, it should be clarified what confirmed (with references), what changed and what added, in way to have a complete description of the species (and in English; in several cases the original description is in Italian). If this cannot be done in Results, it could be done later in a taxonomic account.
3. A species cannot be defined parthenogenetic only because no males are found. In this case parthenogenesis and thelytoky are deductions and no data. The data is the absence of males. Parthenogenesis is highly probable only when a population with a significant number of specimens is found. If so, it must be specified.

Specific comments

Page 3.44: 2018b instead of 2018

Page 3.51: 2019 instead of 2019a. There is only one reference for that year.

Page 4.66: after 'key' add 'of'

Page 4.80: Dastych, 1979a

Page 4.84: I don't understand where these slides are mentioned in the results.

Page 4.88: after 'and' add ', together with the material cited above,'

Page 5.98: micrometers

Page 5.106: 2017 a or b? 2019 without 'a'. There is only one paper in that year.

Page 6.133: it is not clear why in this analysis the two 18S and 28S partitions have different substitution models with respect to those found above. Has the alignment changed? Were new sequences added? If so, please specify. If not, please consider carefully your analysis.

Page 6.149: It is a lower value than that reported in line 172

Page 7.172: Topology is not sufficiently supported. It is better not to take into account the values below 70% and therefore it would also be better not to highlight them. Furthermore, in the caption of figure 1B you write of unsupported values when they are below 60%, while in M&M when they are below 55%.

Page 7.175: BI and not Bi

Page 7.180: see General comments.

Page 8.206: What means ps? It is not explained either in the text and in the references Kristensen (1987) and Gąsiorek et al. (2017, 2019)

Page 12.312: Certainly the type of parthenogenesis was and has not yet been studied. Although probable, thelytoky cannot be confirmed, but only supposed. More likely if the number of specimens and populations considered is high, but the number is not specified here. To avoid misunderstandings, it is better to write: "In *C. holmeni* only females have been found".

Page 17.480: a or b?

Page 19.539: In the two figures of this work *Mopsechiniscus* is no longer related to *Antechiniscus* + *Cornechiniscus* than to *Pseudechiniscus*

Page 19.541: Guidetti et al. (2019) is lacking in references

Page 19.563: Miller et al. is quoted as 2012 in references

Page 20.568: a or b?

Page 20.570-573: not necessary speculation

Page 20.593: a or b?

Page 20.595: Miller et al. is 2012 in references

Page 20.596: a or b?

Page 20.608-609: What means? Moreover, it is better to write: "in *Acanthechiniscus* and *Cornechiniscus* only females were found"

Page 23.693: *Cryoconicus* has been recently found in Antarctica (Guidetti et al., 2019)

Page 24.728: Only females, instead of parthenogenetic. The presence of females is a data, parthenogenesis is a deduction, even though very probable.

Page 25.747: only females

Some confusion with the caption of figure 16: it falls in the following page. Avoid the same thing happening in the final draft.

Review form: Reviewer 2 (Reinhardt Kristensen)

Is the manuscript scientifically sound in its present form?

Yes

Are the interpretations and conclusions justified by the results?

Yes

Is the language acceptable?

Yes

Do you have any ethical concerns with this paper?

No

Have you any concerns about statistical analyses in this paper?

No

Recommendation?

Accept with minor revision (please list in comments)

Comments to the Author(s)

Please see the attached file (Appendix A).

Decision letter (RSOS-200581.R0)

28-Apr-2020

Dear Mr Gąsiorek

On behalf of the Editors, I am pleased to inform you that your Manuscript RSOS-200581 entitled "Revised *Cornechiniscus* (Heterotardigrada) and new phylogenetic analyses negate the recently hypothesised echiniscid subfamilies and tribes" has been accepted for publication in Royal Society Open Science subject to minor revision in accordance with the referee suggestions. Please find the referees' comments at the end of this email.

The reviewers and handling editors have recommended publication, but also suggest some minor revisions to your manuscript. Therefore, I invite you to respond to the comments and revise your manuscript.

- Ethics statement

- Data accessibility

<http://datadryad.org/submit?journalID=RSOS&manu=RSOS-200581>

- Competing interests

- Authors' contributions

- Acknowledgements

- Funding statement

Because the schedule for publication is very tight, it is a condition of publication that you submit the revised version of your manuscript before 07-May-2020. Please note that the revision deadline will expire at 00.00am on this date. If you do not think you will be able to meet this date please let me know immediately.

If your manuscript is newly submitted and subsequently accepted for publication, you will be asked to pay the article processing charge, unless you request a waiver and this is approved by

Royal Society Publishing. You can find out more about the charges at <https://royalsocietypublishing.org/rsos/charges>. Should you have any queries, please contact openscience@royalsociety.org.

on behalf of Dr Maximilian Telford (Associate Editor) and Kevin Padian (Subject Editor)
openscience@royalsociety.org

Associate Editor Comments to Author (Dr Maximilian Telford):
Comments to the Author:

The reviewers are very supportive of publication but both have a list of required corrections. The m.s can be accepted with minor revisions but these revisions must be undertaken carefully. Please ensure the revised version has been very carefully read by a native English speaker to ensure the highest quality.

Reviewer comments to Author:
Reviewer: 1

Comments to the Author(s)
General comments

The work is well written, the critical part on previous phylogeny data is well supported and should be accepted, the descriptions are accurate, there is a lot of morphometry and the images are particularly beautiful and demonstrative.

My suggestion is for minor changes:

1. Topology of Fig. 1B is not sufficiently supported and could create misunderstanding. My suggestion is to delete the values under 70%.
2. In the description of the amendments a reference to the original description is lacking. As a consequence, it is not clear what was contained in the original description, what added and what changed. In several cases the original description is well done, not in other cases. Considering that the authors have examined the type material of several species, it should be clarified what confirmed (with references), what changed and what added, in way to have a complete description of the species (and in English; in several cases the original description is in Italian). If this cannot done in Results, it could be done later in a taxonomic account.
3. A species cannot be defined parthenogenetic only because no males are found. In this case parthenogenesis and thelytoky are deductions and no data. The data is the absence of males. Parthenogenesis is highly probable only when a population with a significant number of specimens is found. If so, it must be specified.

Specific comments

Page 3.44: 2018b instead of 2018

Page 3.51: 2019 instead of 2019a. There is only one reference for that year.

Page 4.66: after 'key' add 'of'

Page 4.80: Dastych, 1979a

Page 4.84: I don't understand where these slides are mentioned in the results.

Page 4.88: after 'and' add ', together with the material cited above,'

Page 5.98: micrometers

Page 5.106: 2017 a or b? 2019 without 'a'. There is only one paper in that year.

Page 6.133: it is not clear why in this analysis the two 18S and 28S partitions have different substitution models with respect to those found above. Has the alignment changed? Were new sequences added? If so, please specify. If not, please consider carefully your analysis.

Page 6.149: It is a lower value than that reported in line 172

Page 7.172: Topology is not sufficiently supported. It is better not to take into account the values below 70% and therefore it would also be better not to highlight them. Furthermore, in the caption of figure 1B you write of unsupported values when they are below 60%, while in M&M when they are below 55%.

Page 7.175: BI and not Bi

Page 7.180: see General comments.

Page 8.206: What means ps? It is not explained either in the text and in the references Kristensen (1987) and Gąsiorek et al. (2017, 2019)

Page 12.312: Certainly the type of parthenogenesis was and has not yet been studied. Although probable, thelytoky cannot be confirmed, but only supposed. More likely if the number of specimens and populations considered is high, but the number is not specified here. To avoid misunderstandings, it is better to write: "In *C. holmeni* only females have been found".

Page 17.480: a or b?

Page 19.539: In the two figures of this work *Mopsechiniscus* is no longer related to *Antechiniscus* + *Cornechiniscus* than to *Pseudechiniscus*

Page 19.541: Guidetti et al. (2019) is lacking in references

Page 19.563: Miller et al. is quoted as 2012 in references

Page 20.568: a or b?

Page 20.570-573: not necessary speculation

Page 20.593: a or b?

Page 20.595: Miller et al. is 2012 in references

Page 20.596: a or b?

Page 20.608-609: What means? Moreover, it is better to write: "in *Acanthechiniscus* and *Cornechiniscus* only females were found"

Page 23.693: *Cryoconicus* has been recently found in Antarctica (Guidetti et al., 2019)

Page 24.728: Only females, instead of parthenogenetic. The presence of females is a data, parthenogenesis is a deduction, even though very probable.

Page 25.747: only females

Some confusion with the caption of figure 16: it falls in the following page. Avoid the same thing happening in the final draft.

Reviewer: 2

Comments to the Author(s)

Please see the attached file - Review

Author's Response to Decision Letter for (RSOS-200581.R0)

See Appendix B.

Decision letter (RSOS-200581.R1)

12-May-2020

Dear Mr Gąsiorek,

It is a pleasure to accept your manuscript entitled "Revised Cornechiniscus (Heterotardigrada) and new phylogenetic analyses negate echiniscid subfamilies and tribes" in its current form for publication in Royal Society Open Science. The comments of the reviewer(s) who reviewed your manuscript are included at the foot of this letter.

on behalf of Dr Maximilian Telford (Associate Editor) and Kevin Padian (Subject Editor)
openscience@royalsociety.org

Associate Editor Comments to Author (Dr Maximilian Telford):

The authors have dealt thoroughly with the comments from the two reviewers (who very much approved of most of the initial submission).

Appendix A

Manuscript Review 184. – Manuscript ID RSOS-200581_ Royal Society Open Science

Title: Revised *Cornechiniscus* (Heterotardigrada) and new phylogenetic analyses negate the recently hypothesized echiniscid subfamilies and tribes.

Authors: PIOTR GAŚIOREK & ŁUKASZ MICHALCZYK

GENERAL. The manuscript by Gašiorek & Michalczyk is outstanding. It combines the classical way of tardigrade research - morphology and systematic - done with light microscopy (LM), scanning electron microscopy (SEM) and with a very sophisticated molecular dataset of the rare genus *Cornechiniscus* and related genera (see, Fig. 1A and Fig. 1B). The most significant is that the authors have observed all the eighth species of *Cornechiniscus* there are described today and illustrated them all nicely in the manuscript! That is really unique! I think that the authors have to check the “References” and citations very carefully again, especially the authors own “References”. However, the list of “References” for the genus *Cornechiniscus* is really fantastic; perhaps the authors lack a few reports from Arctic. The last part of the manuscript is dealing with the recently published paper (Guil et al., 2019) about “An upgraded comprehensive multilocus phylogeny of the Tardigrada tree of life”. This is really great that the authors “negate the recently hypothesized echiniscid subfamily and tribes”, because Guil et al. 2019 nearly only have based the phylogeny of Tardigrada by morphology. However, the author’s Table 12 give us not any new resolutions of “tribal classification of the family Echiniscidae”. It is really great that the authors focus on the problem with the classification of the family Echiniscidae; but, how can the authors know that the Parechiniscinae is paraphyletic, when we not have published the molecular data of *Novechiniscus* and all the new species of *Parechiniscus*? Finally, both authors are the world experts in the genus *Bryodelphax* (Bryodelphaxini). It has been a great pleasure to read all their papers. However, why are the problem with genus *Bryochoerus*

(sister genus to *Bryodelphax*, see Kristensen 1987), not mentioned in the manuscript and in Table 12? So many new species of *Bryochoerus* are under descriptions and the new species from China - have already been described (please cite also the papers about *Bryochoerus* – in the manuscript) – also when the authors think that the genus *Bryochoerus* Marcus, 1936 is very dubiously (Marcus, 1936; Thulin, unpubl.; Kristensen, 1987).

There are a very few numbers of minor editorial corrections I have comment below, together with a few points that have to be discussed and corrected before the paper can be published with **a minor revision**. It is a fantastic manuscript with lot of new information about the whole phylum Tardigrada.

1). **Title:** I would like to shorting the title – it is very long now and why to focus of a single (**stupid**) paper (Guil, Jørgensen and Kristensen, 2019)? Please change the title to: **Revised *Cornechiniscus* (Echiniscidae, Heterotardigrada) and new phylogenetic analyses of echiniscid subfamilies and tribes.**

2). **Keywords:** I would like to include the word **limno-terrestrial life cycle**. Echiniscidae can now be deleted, when it is included in the title.

3). **Running title:** Phylogeny of *Cornechiniscus*. **Great running title.**

4). The reviewer really lacks all information about what species of *Cornechiniscus* have been investigated “outside” the author’s own collections (**great** Table 1). Please, make a Table (“2”) with all species of *Cornechiniscus* used and illustrated in this manuscript. I know it is very difficult to get all the information as in Table 1; however, we need “Locality” and “Collector” of all eight species of *Cornechiniscus*, because they all have been illustrated so nicely in the manuscript. **However, where are the specimens coming from!!**

5). **References.**

The authors cite correctly that *Cornechiniscus holmeni* original was described from Greenland (Petersen, 1951) and they mentioned all the other localities, where this species have been found. However, in the Table 3 they also mention Igloolik (Turton Bay). Please, cite therefore the Igloolik paper:

Jørgensen, M. and Kristensen, R. M. Meiofauna investigations from Igloolik, N.W.T. Arctic Canada. Biology Course 1989, Igloolik N.W.T. Canada (ed. M. Jørgensen). University of Copenhagen p.p. 61-80.

Please, cite also the newest tardigrade papers of :

Jørgensen, A. Kristensen, R.M. and Møbjerg, N. (2019). Chapter 3. Phylogeny and Integrative Taxonomy of Tardigrada. R.O. Schill (ed.) Water Bears: The Biology of Tardigrades. Zoological Monographs 2, pp.95- 114. Please see Figs 3.5 and 3.6.

Møbjerg N., Jørgensen, A., Kristensen, R.M. and Neves, R.C. (2019). Chapter 2.

Morphology and Functional Anatomy. R.O. Schill (ed.) Water Bears: The Biology of Tardigrades. Zoological Monographs 2, pp.57-94.

Concerning the problem with the genus *Bryochoerus* - I think the authors should cite the relative new description of the Chinese species:

Xue, J., Li, X., Wang, L., Xian, P. and Chen, H. (2017). *Bryochoerus liupanensis* sp. nov. and *Pseudechiniscus chengi* sp. nov. (Tardigrada: Heterotardigrada: Echiniscidae) from China. Zootaxa 4291(2): 324-334.

6). **Figures and Tables.** This is the most fantastic part of the whole manuscript. Both authors are known to make great SEM-illustrations of new species of tardigrades, but how can they

use very old museum material in this unique way? Please, see figure 4 of *Cornechiniscus holmeni* from Greenland and Tables 3 and 4. A small correction: there are 3 localities in Greenland called Âta – therefore, please write Âta, Nuussuaq (Greenland).

Fig. 1A and 1B. These two figures are what we all are waiting for: The phylogenetic relationships in the family Echiniscidae. We all use *Oreella mollis* as an out-group! Perhaps, this is not the best terrestrial heterotardigrade to use. Could we also use the tidal species *Echiniscoides sigismundi* (family Echiniscoididae) as an out-group - too? The genus *Oreella* in the family Oreellidae is very interesting because the females have seminal receptacles similar to the females in the order Arthrotardigrada. Seminal receptacles are never seen in the Echiniscidae and Echiniscoididae.

7). I am not the best to criticized English language and I cannot help in this way, but it has been a great pleasure to read the manuscript; however, I think that the manuscript need a reviewer to correct the English – a reviewer who is much better to English than me.

8). Small editorial corrections:

Page 9, line 245 merged with rigid the *flagellum* – are the authors meaning merged with the rigid *flagellum*?

Please check all citations both in text and in the “References” very carefully, especially the German’s references.

In the “Key to *Cornechiniscus* spp.” – please, use the same abbreviation for diameter of the endocuticular pillars (\emptyset) in line 742 (\emptyset) and line 744 (\emptyset), page 25.

9). Acknowledgements. Please, acknowledge Asger Ken Pedersen (Museum of Natural History of Denmark, University of Copenhagen) for collecting the moss samples containing *Cornechiniscus madagascariensis* from Ethiopia.

10). It has been a great pleasure to read the manuscript and the manuscript should be

published with any doubt in “Royal Society Open Science”. **In fact the manuscript is outstanding.** However; I will recommend **a minor revision** with the very few points I have written in the review and with the citations of some of the last publications.

Appendix B

Dear Editors,

We are very grateful for the reviews and your conditional acceptance of our manuscript. We agreed with the vast majority of remarks and we provided our detailed response to the few points we did not agree with. As requested, the revised version of the manuscript was proof-read by a native English speaker.

Yours sincerely,

Piotr Gąsiorek & Łukasz Michalczyk

Reviewer #1

General comments

The work is well written, the critical part on previous phylogeny data is well supported and should be accepted, the descriptions are accurate, there is a lot of morphometry and the images are particularly beautiful and demonstrative.

Thank you for your kind words and suggestions, they are all addressed below.

My suggestion is for minor changes:

1. Topology of Fig. 1B is not sufficiently supported and could create misunderstanding. My suggestion is to delete the values under 70%.

Changed as suggested.

2. In the description of the amendments a reference to the original description is lacking. As a consequence, it is not clear what was contained in the original description, what added and what changed. In several cases the original description is well done, not in other cases. Considering that the authors have examined the type material of several species, it should be clarified what confirmed (with references), what changed and what added, in way to have a complete description of the species (and in English; in several cases the original description is in Italian). If this cannot done in Results, it could be done later in a taxonomic account.

Changed as suggested.

3. A species cannot be defined parthenogenetic only because no males are found. In this case parthenogenesis and thelytoky are deductions and no data. The data is the absence of males. Parthenogenesis is highly probable only when a population with a significant number of specimens is found. If so, it must be specified.

Changed as suggested.

Specific comments

Page 3.44: 2018b instead of 2018

Page 3.51: 2019 instead of 2019a. There is only one reference for that year.

Page 5.106: 2017 a or b? 2019 without 'a'. There is only one paper in that year.

Page 17.480: a or b?

Page 20.593: a or b?

Page 20.596: a or b?

Not needed to specify after adjusting the references to the RSOS requirements.

Page 4.66: after 'key' add 'of'

Added.

Page 4.80: Dastych, 1979a

Not needed to specify after adjusting the references to the RSOS requirements.

Page 4.84: I don't understand where these slides are mentioned in the results.

We mention them in the Discussion as new identifications, e.g.: “However, the examination of slides deposited in Copenhagen revealed its presence also in south-western Kenya (Gembe Hills, expedition 1987–1988), northern Tanzania, Ethiopia (expedition 2016), and Kashmir (see also above), which means that the distribution of *C. madagascariensis* may cover the entire Indian Ocean basin.”

Page 4.88: after ‘and’ add ‘, together with the material cited above,’

Added.

Page 5.98: micrometers

We prefer to keep the British English spelling (micrometres).

Page 6.133: it is not clear why in this analysis the two 18S and 28S partitions have different substitution models with respect to those found above. Has the alignment changed? Were new sequences added? If so, please specify. If not, please consider carefully your analysis.

Specified as requested.

Page 6.149: It is a lower value than that reported in line 172

Changed consistently into 70% and the figure was modified accordingly.

Page 7.172: Topology is not sufficiently supported. It is better not to take into account the values below 70% and therefore it would also be better not to highlight them. Furthermore, in the caption of figure 1B you write of unsupported values when they are below 60%, while in M&M when they are below 55%.

Changed consistently into 70% and the figure was modified accordingly.

Page 7.175: BI and not Bi

Corrected.

Page 7.180: see General comments.

The response is provided above.

Page 8.206: What means ps? It is not explained either in the text and in the references Kristensen (1987) and Gąsiorek et al. (2017, 2019)

Specified now in the M&M: “(...) “ps” signifies appendages present on the posterior margin of the pseudosegmental plate IV’.”

Page 12.312: Certainly the type of parthenogenesis was and has not yet been studied. Although probable, thelytoky cannot be confirmed, but only supposed. More likely if the number of specimens and populations

considered is high, but the number is not specified here. To avoid misunderstandings, it is better to write: "In *C. holmeni* only females have been found".

Corrected.

Page 19.539: In the two figures of this work *Mopsechiniscus* is no longer related to *Antechiniscus* + *Cornechiniscus* than to *Pseudechiniscus*

We discussed both lines of evidence: molecular and morphological, resulting in contrasting results. Thus, the current sentence is correct.

Page 19.541: Guidetti et al. (2019) is lacking in references

Corrected (Cesari et al. 2020 is the right reference).

Page 19.563: Miller et al. is quoted as 2012 in references

Corrected.

Page 20.568: a or b?

Not needed to specify after adjusting the references to the RSOS requirements.

Page 20.570-573: not necessary speculation

This is supported by the currently available morphological phylogenies (Jørgensen 2000, Gąsiorek et al. 2018) but we changed the word "ignored" to "did not take into consideration" to avoid the feeling of speculation.

Page 20.595: Miller et al. is 2012 in references

Corrected.

Page 20.608-609: What means? Moreover, it is better to write: "in *Acanthechiniscus* and *Cornechiniscus* only females were found"

Specified.

Page 23.693: *Cryoconicus* has been recently found in Antarctica (Guidetti et al., 2019)

Citation added.

Page 24.728: Only females, instead of parthenogenetic. The presence of females is a data, parthenogenesis is a deduction, even though very probable.

Changed accordingly.

Page 25.747: only females

Changed accordingly.

Some confusion with the caption of figure 16: it falls in the following page. Avoid the same thing happening in the final draft.

Figures in the original submission were in a PDF format, but now they will be uploaded as TIFF files, so this will not be a problem anymore.

Reviewer #2

GENERAL. The manuscript by Gąsiorek & Michalczyk is outstanding. It combines the classical way of tardigrade research - morphology and systematic - done with light microscopy (LM), scanning electron microscopy (SEM) and with a very sophisticated molecular dataset of the rare genus *Cornechiniscus* and related genera (see, Fig. 1A and Fig. 1B). The most significant is that the authors have observed all the eighth species of *Cornechiniscus* there are described today and illustrated them all nicely in the manuscript! That is really unique! I think that the authors have to check the “References” and citations very carefully again, especially the authors own “References”. However, the list of “References” for the genus *Cornechiniscus* is really fantastic; perhaps the authors lack a few reports from Arctic. The last part of the manuscript is dealing with the recently published paper (Guil et al., 2019) about “An upgraded comprehensive multilocus phylogeny of the Tardigrada tree of life”. This is really great that the authors “negate the recently hypothesized echiniscid subfamily and tribes”, because Guil et al. 2019 nearly only have based the phylogeny of Tardigrada by morphology. However, the author’s Table 12 give us not any new resolutions of “tribal classification of the family Echiniscidae”. It is really great that the authors focus on the problem with the classification of the family Echiniscidae; but, how can the authors know that the Parechiniscinae is paraphyletic, when we not have published the molecular data of *Novechiniscus* and all the new species of *Parechiniscus*? Finally, both authors are the world experts in the genus *Bryodelphax* (Bryodelphaxini). It has been a great pleasure to read all their papers. However, why are the problem with genus *Bryochoerus* (sister genus to *Bryodelphax*, see Kristensen 1987), not mentioned in the manuscript and in Table 12? So many new species of *Bryochoerus* are under descriptions and the new species from China - have already been described (please cite also the papers about *Bryochoerus* – in the manuscript) – also when the authors think that the genus *Bryochoerus* Marcus, 1936 is very dubiously (Marcus, 1936; Thulin, unpubl.; Kristensen, 1987). There are a very few numbers of minor editorial corrections I have comment below, together with a few points that have to be discussed and corrected before the paper can be published with a minor revision. It is a fantastic manuscript with lot of new information about the whole phylum Tardigrada.

Thank you for your kind words and remarks – the detailed response is provided below. We double-checked the References. Table 12 summarises the drawbacks of Guil et al.’s systematic units. We clearly stated in the Discussion that, in our opinion and considering all available data, no subfamilies and, maybe, tribes, can be introduced for this family at present.

It is true that there are no available genetic data for *Novechiniscus*, but already at the time of publishing Guil et al. were there morphological phylogenies accessible (Jørgensen 2000, Gąsiorek et al. 2018) – all denying monophyletic character of the putative subfamily “Parechiniscinae”. We clearly stated in the Discussion that these phylogenies were neglected by Guil et al. We do not know why (especially as the first work was performed by one of the co-authors of that unfortunate paper), but it non-deliberately gives an impression of cherry-picking which data the researchers like or not.

Finally, we did not elaborate on the *Bryodelphax-Bryochoerus* problem as it is a side issue, outside the scope of our manuscript. Nevertheless, we added additional citations signalling this conundrum.

1). Title: I would like to shorting the title – it is very long now and why to focus of a single (stupid) paper (Guil, Jørgensen and Kristensen, 2019)? Please change the title to: Revised *Cornechiniscus* (Echiniscidae, Heterotardigrada) and new phylogenetic analyses of echiniscid subfamilies and tribes.

We changed the title slightly “Revised *Cornechiniscus* (Heterotardigrada) and new phylogenetic analyses negate echiniscid subfamilies and tribes”, but we cannot delete the second part of the title altogether, as the bulk of the Discussion addresses the vague analyses of Guil et al., and the present title reflects the content of

the manuscript. Moreover, we did not analyse “subfamilies and tribes”, as our a priori assumption was that such a grouping is artificial.

2). Keywords: I would like to include the word limno-terrestrial life cycle. Echiniscidae can now be deleted, when it is included in the title.

“Echiniscidae” deleted, but the proposed terms has barely anything to do with the issues talked in this manuscript. “phylogenetic congruence” added.

3). Running title: Phylogeny of *Cornechiniscus*. Great running title.

Thank you.

4). The reviewer really lacks all information about what species of *Cornechiniscus* have been investigated “outside” the author’s own collections (great Table 1). Please, make a Table (“2”) with all species of *Cornechiniscus* used and illustrated in this manuscript. I know it is very difficult to get all the information as in Table 1; however, we need “Locality” and “Collector” of all eight species of *Cornechiniscus*, because they all have been illustrated so nicely in the manuscript. However, where are the specimens coming from!!

Figure legends provide the information on where specimens were collected directly or indirectly by stating that they represent a type series (holotype/paratype etc.) for which collection data are provided in Table 1. We also clearly stated in the M&M section that all type series were examined, therefore a reader should refer to original descriptions in such cases. Thus, we think that the origin of populations is well-explained in the manuscript and does not require an addition of a new table.

5). References.

The authors cite correctly that *Cornechiniscus holmeni* original was described from Greenland (Petersen, 1951) and they mentioned all the other localities, where this species have been found. However, in the Table 3 they also mention Igloodik (Turton Bay). Please, cite therefore the Igloodik paper:

Jørgensen, M. and Kristensen, R. M. Meiofauna investigations from Igloodik, N.W.T. Arctic Canada. Biology Course 1989, Igloodik N.W.T. Canada (ed. M. Jørgensen). University of Copenhagen p.p. 61-80.

Cited.

Please, cite also the newest tardigrade papers of :

Jørgensen, A. Kristensen, R.M. and Møbjerg, N. (2019). Chapter 3. Phylogeny and Integrative Taxonomy of Tardigrada. R.O. Schill (ed.) Water Bears: The Biology of Tardigrades. Zoological Monographs 2, pp.95- 114. Please see Figs 3.5 and 3.6.

Cited.

Møbjerg N., Jørgensen, A., Kristensen, R.M. and Neves, R.C. (2019). Chapter 2. Morphology and Functional Anatomy. R.O. Schill (ed.) Water Bears: The Biology of Tardigrades. Zoological Monographs 2, pp.57-94.

Cited.

Concerning the problem with the genus *Brychoerus* - I think the authors should cite the relative new description of the Chinese species:

Xue, J., Li, X., Wang, L., Xian, P. and Chen, H. (2017). *Bryochoerus liupanensis* sp. nov. and *Pseudechiniscus chengi* sp. nov. (Tardigrada: Heterotardigrada: Echiniscidae) from China. *Zootaxa* 4291(2): 324-334.

Cited.

6). Figures and Tables. This is the most fantastic part of the whole manuscript. Both authors are known to make great SEM-illustrations of new species of tardigrades, but how can they use very old museum material in this unique way? Please, see figure 4 of *Cornechiniscus holmeni* from Greenland and Tables 3 and 4. A small correction: there are 3 localities in Greenland called Âta – therefore, please write Âta, Nuussuaq (Greenland).

Corrected.

Fig. 1A and 1B. These two figures are what we all are waiting for: The phylogenetic relationships in the family Echiniscidae. We all use *Oreella mollis* as an out-group! Perhaps, this is not the best terrestrial heterotardigrade to use. Could we also use the tidal species *Echiniscoides sigismundi* (family Echiniscoididae) as an out-group - too? The genus *Oreella* in the family Oreellidae is very interesting because the females have seminal receptacles similar to the females in the order Arthrotardigrada. Seminal receptacles are never seen in the Echiniscidae and Echiniscoididae.

Phylogenies with other outgroups were provided as supplementary materials.

7). I am not the best to criticize English language and I cannot help in this way, but it has been a great pleasure to read the manuscript; however, I think that the manuscript needs a reviewer to correct the English – a reviewer who is much better at English than me.

The manuscript has now been proof-read by an English native speaker.

8). Small editorial corrections:

Page 9, line 245 merged with rigid the flagellum – are the authors meaning merged with the rigid flagellum?

Corrected.

Please check all citations both in text and in the “References” very carefully, especially the German’s references.

Double-checked and adjusted to the RSOS requirements.

In the “Key to *Cornechiniscus* spp.” – please, use the same abbreviation for diameter of the endocuticular pillars (\emptyset) in line 742 (\emptyset) and line 744 (\emptyset), page 25.

Corrected.

9). Acknowledgements. Please, acknowledge Asger Ken Pedersen (Museum of Natural History of Denmark, University of Copenhagen) for collecting the moss samples containing *Cornechiniscus madagascariensis* from Ethiopia.

Added.

10). It has been a great pleasure to read the manuscript and the manuscript should be published with any doubt in “Royal Society Open Science”. In fact the manuscript is outstanding. However; I will recommend a

minor revision with the very few points I have written in the review and with the citations of some of the last publications.

Thank you once again for your detailed and encouraging review.